



# Cold-water corals and hydrocarbon-rich seepage in the
# Pompeia Province (Gulf of Cádiz) — living on the edge
Blanca Rincón-Tomás[1], Jan-Peter Duda[2,3], Luis Somoza[4], Javier González[4], Dominik
Schneider[1], Teresa Medialdea[4], Pedro Madureira[5], Michael Hoppert[1], and Joachim Reitner[2,3]
[1]Georg-August-University Göttingen, Institute of Microbiology and Genetics, Grisebachstraße 8, 37077
Göttingen, Germany
[2]Georg-August-University Göttingen, Göttingen Centre of Geosciences, Goldschmidtstraße 3, 37077 Göttingen,
Germany
[3]Göttingen Academy of Sciences and Humanities, Theaterstraße 7, 37073 Göttingen, Germany
[4]Marine Geology Dept., Geological Survey of Spain, IGME, Ríos Rosas 23, 28003 Madrid, Spain
[5]Estrutura de Missão para a Extensão da Plataforma Continental. Rua Costa Pinto 165, 2770-047 Paço de Arcos,
Portugal
*Correspondence to*: Blanca Rincón-Tomás (b.rincontomas@gmail.com)
**Abstract**. Azooxanthellate cold-water corals (CWCs) are globally widespread and have commonly been found in
areas of active fluid seepage. The relationship between the CWCs and these fluids, however, is not well understood.
This study aims at unravelling the relationship between CWC development and hydrocarbon-rich seepage in the
Pompeia Province (Gulf of Cádiz, Atlantic Ocean). This region comprises mud volcanoes, coral ridges and fields
of coral mounds, which are all affected by the tectonically driven seepage of hydrocarbon-rich fluids. Rate and
type of seepage (i.e. focused, scattered, diffused, eruptive), however, is tightly controlled by a complex system of
faults and diapirs. Early diagenetic carbonates from the currently active Al Gacel MV exhibit $\delta^{13}$C-signatures
down to −28.77 ‰ VPDB, indicating biologically derived methane as the main carbon source. The same samples
contained $^{13}$C-depleted lipid biomarkers diagnostic for archaea such as crocetane ($\delta^{13}$C down to −101.2 ‰ VPDB)
and PMI ($\delta^{13}$C down to −102.9 ‰ VPDB), evidencing microbially mediated anaerobic oxidation of methane
(AOM). This is further supported by next generation DNA sequencing data, demonstrating the presence of AOM
related microorganisms (ANME archaea, sulfate-reducing bacteria) in the carbonate. Embedded corals in some of
the carbonates and CWC fragments exhibit less negative $\delta^{13}$C values (−8.08 to −1.39‰ VPDB), pointing against
the use of methane as carbon source. Likewise, the absence of DNA from methane- and sulfide-oxidizing microbes
in a sampled coral does not support a chemosynthetic lifestyle of these organisms. In the light of these findings, it
appears that the CWCs benefit rather indirectly from hydrocarbon-rich seepage by using methane-derived
authigenic carbonates as substratum for colonization. At the same time, chemosynthetic organisms at active sites
prevent coral dissolution and necrosis by feeding on the seeped fluids (i. e. methane, sulfate, hydrogen sulfide),
allowing cold-water corals to colonize carbonates currently affected by hydrocarbon-rich seepage.
## 1. Introduction
Cold-water corals (CWCs) are a widespread, non-phylogenetic group of cnidarians which include hard skeleton
scleractinian corals, soft-tissue octocorals, gold corals, black corals and hydrocorals (Roberts et al., 2006; Roberts
et al., 2009; Cordes et al., 2016). They typically thrive at low temperatures (4 − 12 ℃) and occur in water depths
of ca. 50 − 4000 m. CWCs are azooxanthellate and solely rely on their nutrition as energy and carbon sources
(Roberts et al., 2009). Some scleractinian corals (e.g. *Lophelia pertusa, Madrepora oculata, Dendrophyllia*



*cornigera, Dendrophyllia alternata, Eguchipsammia cornucopia*) are able to form colonies or even large carbonate
mounds (Rogers et al., 1999; Wienberg et al., 2009; Watling et al., 2011; Somoza et al., 2014). Large vertical
mounds and elongated ridges formed by episodic growth of scleractinian corals (mainly *Lophelia pertusa*) are for
instance widely distributed along the continental margins of the Atlantic Ocean (Roberts et al., 2009). These
systems are of great ecological value since they offer sites for resting-, breeding-, and feeding for various
invertebrates and fishes (Cordes et al., 2016 and references therein).
Several ecological forces are discussed to control the initial settling, growth, and decline of CWCs. These include,
among others, an availability of suitable substrates for coral larvae settlement, low sedimentation rates,
oceanographic boundary conditions (e.g. salinity, temperature and density of the ocean water) and a sufficient
supply of nutrients through topographically controlled currents systems (e.g. Freiwald et al., 1999, 2002;
Mortensen et al., 2001; Roberts et al., 2003; Thiem et al., 2006; Dorschel et al., 2007; Dullo et al., 2008; Frank et
al., 2011; Van Rooij et al., 2011; Hebbeln et al., 2016). Alternatively, CWC ecosystems may be directly fueled
by fluid seepage, providing a source of e.g. sulfur compounds, nitrogen compounds, P, $CO_2$ and/or hydrocarbons
(Hovland, 1990; Hovland and Thomsen, 1997; Hovland et al., 1998). This relationship is supported by the common
co-occurrence of CWC-mounds and hydrocarbon-rich seeps around the world as e.g. at the Hikurangi Margin in
New Zealand (Liebetrau et al., 2010), the Brazil margin (e.g. Gomes-Sumida et al., 2004), the Darwin Mounds in
the northern Rockall Trough (Huvenne et al., 2009), the Kristin field on the Norwegian shelf (Hovland et al.,
2012), the western Alborán Sea (Margreth et al., 2011), and the Gulf of Cádiz (e.g. Díaz-del-Río et al., 2003;
Foubert et al., 2008). However, CWCs may also benefit rather indirectly from seepage. For instance, methane-
derived authigenic carbonates (MDACs) formed through the microbially mediated anaerobic oxidation of methane
(AOM; Suess & Whiticar, 1989; Hinrichs et al., 1999; Thiel et al., 1999; Boetius et al., 2000; Hinrichs & Boetius,
2002; Valentine, 2002, Boetius & Suess, 2004) potentially provide hard substrata for larval settlement (e.g. Díaz-
del-Rio et al., 2003; Van Rooij et al., 2011; Magalhães et al. 2012; Le Bris et al., 2016; Rueda et al., 2016). On the
other hand, larger hydrocarbon-rich seepage related structures such as mud volcanoes and carbonate mud mounds
act as morphological barriers favoring turbulent water currents that deliver nutrients to the corals (Roberts et al.,
2009; Wienberg et al., 2009; Margreth et al., 2011; Vandorpe et al., 2016).
In the Gulf of Cádiz, most CWC occurrences are "coral graveyards" (i.e., with only few living corals) that are
situated along the Iberian and Moroccan margins. These CWC systems are typically associated with diapiric
ridges, steep fault-controlled escarpments, and mud volcanoes (MVs) such as the Faro MV, Hesperides MV,
Mekness MV, and MVs in the Pen Duick Mud Volcano Province (Foubert et al., 2008; Wienberg et al., 2009).
MVs (and other conspicuous morphological structures in this region such as pockmarks) are formed through
tectonically induced fluid flow (Pinheiro et al., 2003; Somoza et al., 2003; Medialdea et al., 2009; León et al.,
2010; 2012). This is because of the high regional tectonic activity and high fluid contents of sediments in this area
(mainly $CH_4$ and, to a lesser extent, $H_2S$, $CO_2$, and $N_2$: Pinheiro et al., 2003; Hensen et al., 2007; Scholz et al.,
2009; Smith et al., 2010; González et al., 2012). However, the exact influence of fluid flow on CWC growth in
this region remains elusive.
This study aims at elucidating the linkage between the present-day formation of MDACs and CWCs development
along the Pompeia Province (**Fig. 1**), which englobes mud volcanoes as the Al Gacel MV (León et al., 2012),
diapiric coral ridges and mounds. We address this question by the combined analysis of high-resolution ROV
underwater images, geophysical data (e.g. seabed topography, deep high-resolution multichannel seismic
reflection data), and sample materials (petrographic features, $\delta^{13}$C- and $\delta^{18}$O-signatures of carbonates, lipid



biomarkers and environmental 16s rDNA sequences of the prokaryotic microbial community). Based on our
findings, we propose an integrated model to explain the tempo-spatial and genetic relations between CWCs,
chemosynthetic fauna and hydrocarbon-rich seepage in the study area.

## 2. Materials and Methods

This study is based on collected data from the Pompeia Province, during the Subvent-2 cruise in 2014 aboard the
R/V Sarmiento de Gamboa. The analyzed samples were recovered from the Al Gacel MV (D10-R3, D10-R7, D11-
R8) and the Northern Pompeia Coral Ridge (D03-B1) (**Fig. 1**).

### 2.1. Geophysical survey

Seabed topography of the studied sites was mapped by using an Atlas Hydrosweep DS (15 kHz and 320 beams)
multibeam echosounder (MBES). Simultaneously, ultra-high resolution sub-bottom profiles were acquired with
an Atlas Parasound P-35 parametric chirp profiler ($0.5 - 6$ kHz). Deep high-resolution multichannel seismic
reflection data was obtained using an array of 7 SERCEL gi-guns (system composed of $250 + 150 + 110 + 45$
cubic inches) with a total of 860 cubic inches. The obtained data were recorded with an active streamer
(SIG®16.3x40.175; 150 m length with 3 sections of 40 hydrophones each). The shot interval was 6 seconds and
the recording length 5 seconds two-way travel time (TWT). Data processing (filtering and stacking) was performed
on board with Hot Shots software.

### 2.2. Video survey and analysis

A remotely operated vehicle (ROV-6000 Luso) was used for photographic documentation (high definition digital
camera, 1024x1024 pixel) and sampling. The ROV was further equipped with a STD/CTD-SD204 sensor (*in-situ*
measurements of salinity, temperature, oxygen, conductivity, sound velocity and depth), HydroC$^{TM}$ sensors (*in-*
*situ* measurements of $CO_2$ and $CH_4$) and Niskin bottles ($CH_4$ concentrations).

### 2.3. Petrographic analysis

General petrographic analysis was performed on thin sections (ca. 60 μm thickness) with a Zeiss SteREO
Discovery.V8 stereomicroscope (transmitted- and reflected light) linked to an AxioCam MRc 5-megapixel camera.
Additional detailed petrographic analysis of textural and mineralogical features was conducted on polished thin
sections (ca. 30 μm thickness) using a DM2700P Leica Microscope coupled to a DFC550 digital camera.
Carbonate textures have been classified following Dunham (1962) and Embry & Klovan (1971).

### 2.4. Stable isotopes ($\delta^{13}$C, $\delta^{18}$O) of carbonates

Stable carbon and oxygen isotope measurements were conducted on ca. 0.7 mg carbonate powder obtained with a
high precision drill (ø 0.8 mm). The analyses were performed with a Thermo Scientific Kiel IV carbonate device
coupled to a Finnigan Delta Plus gas isotope mass spectrometer. Reproducibility was checked through the replicate
analysis of a standard (NBS19) and was generally better than 0.1 ‰. Stable carbon and oxygen isotope values are
expressed in the standard $\delta$ notation as per mill (‰) deviations relative to Vienna Pee Dee Belemnite (VPDB).

### 2.5. Lipid biomarker analysis





### 2.5.1. Sample preparation

All materials used were pre-combusted (500 °C for >3 h) and/or extensively rinsed with acetone prior to sample contact. A laboratory blank (pre-combusted sea sand) was prepared and analyzed in parallel to monitor laboratory contaminations.

The preparation and extraction of lipid biomarkers was conducted in orientation to descriptions in Birgel et al. (2006). Briefly, the samples were first carefully crushed with a hammer and internal parts were powdered with a pebble mill (Retsch MM 301, Haan, Germany). Hydrochloric acid (HCl; 10 %) was slowly poured on the powdered samples which were covered with dichloromethane (DCM)-cleaned water. After 24 h of reaction, the residues (pH 3 – 5) were repeatedly washed with water and then lyophilized.

3 g of each residue was saponified with potassium hydroxide (KOH; 6 %) in methanol (MeOH). The residues were then extracted with methanol (40 mL, 2x) and, upon treatment with HCl (10 %) to pH 1, in DCM (40 mL, 2x) by using ultra-sonification. The combined supernatants were partitioned in DCM vs. water (3x). The total organic extracts (TOEs) were dried with sodium sulfate ($NaSO_4$) and evaporated with a gentle stream of $N_2$ to reduce loss of low-boiling compounds (cf. Ahmed and George, 2004).

50 % of each TOE was separated over a silica gel column (0.7 g Merck silica gel 60 conditioned with *n*-hexane; 1.5 cm i.d., 8 cm length) into (a) hydrocarbon (6 mL *n*-hexane), (b) alcohol (7 mL DCM/acetone, 9:1, v:v) and (c) carboxylic acid fractions (DCM/MeOH, 3:1, v:v). Only the hydrocarbons were subjected to gas chromatography–mass spectrometry (GC-MS).

### 2.5.2. Gas chromatography–mass spectrometry (GC-MS)

Lipid biomarker analyses of the hydrocarbon fraction were performed with a Thermo Scientific Trace 1310 GC coupled to a Thermo Scientific Quantum XLS Ultra MS. The GC was equipped with a capillary column (Phenomenex Zebron ZB-5MS, 30 m length, 250 µm inner diameter, 0.25 µm film thickness). Fractions were injected into a splitless injector and transferred to the column at 300 °C. The carrier gas was He at a flow rate of 1.5 mL $min^{-1}$. The GC oven temperature was ramped from 80°C (1 min) to 310 °C at 5 °C $min^{-1}$ (held for 20 min). Electron ionization mass spectra were recorded in full scan mode at an electron energy of 70 eV with a mass range of m/z 50 − 600 and scan time of 0.42 s. Identification of individual compounds was based on comparison of mass spectra and GC retention times with published data and reference compounds.

### 2.5.3 Gas chromatography–combustion–isotope ratio mass spectrometer (GC-C-IRMS)

Compound specific $\delta^{13}C$ analyses were conducted with a Trace GC coupled to a Delta Plus IRMS via a combustion-interface (all Thermo Scientific). The combustion reactor contained CuO, Ni and Pt and was operated at 940°C. The GC was equipped with two serially linked capillary columns (Agilent DB-5 and DB-1; each 30 m length, 250 µm inner diameter, 0.25 µm film thickness). Fractions were injected into a splitless injector and transferred to the GC column at 290°C. The carrier gas was He at a flow rate of 20 ml $min^{-1}$. The temperature program was identical to the one used for GC-MS (see above). $CO_2$ with known $\delta^{13}C$ value was used for internal calibration. Instrument precision was checked using a mixture of *n*-alkanes with known isotopic composition. Carbon isotope ratios are expressed as $\delta^{13}C$ (‰) relative to VPDB.

### 2.6. Amplicon sequencing of 16S rRNA genes



### 2.6.1. DNA extraction and 16S rRNA gene amplification

About $1-4$ g of solid samples were first mashed with mortar and liquid nitrogen to fine powder. Three biological

replicates were used per sample. Total DNA was isolated with a Power Soil DNA Extraction Kit (MO BIO

Laboratories, Carlsbad, CA). All steps were performed according to the manufacturer's instructions.

Bacterial amplicons of the $V3-V4$ region were generated with the primer set MiSeq_Bacteria_V3_forward

primer (5'-TCGTCGGCAGCGTCAGATGTGTATAAGAGACAGCCTACGGGNGGCWGCAG-3') and

MiSeq_Bacteria_V4_reverse primer (5'-

GTCTCGTGGGCTCGGAGATGTGTATAAGAGACAGGACTACHVGGGTATCTAATCC-3'). Likewise,

archaeal amplicons of the $V3-V4$ region were generated with the primer set MiSeq_Archaea_V3_forward primer

(5'-TCGTCGGCAGCGTCAGATGTGTATAAGAGACAG-GGTGBCAGCCGCCGCGGTAA-3') and

MiSeq_Archaea_V4_reverse primer (5'-GTCTCGTGGGCTCGGAGATGTGTATAAGAGACAG-

CCCGCCAATTYCTTTAAG-3'). 50 µl of the PCR reaction mixture for bacterial DNA amplification, contained

1 U Phusion high fidelity DNA polymerase (Biozym Scientific, Oldendorf, Germany), 5% DMSO, 0.2 mM of

each primer, 200 µM dNTP, 0.15 µl of 25 mM $MgCl_2$, and 25 ng of isolated DNA. The PCR protocol for bacterial

DNA amplification included (i) initial denaturation for 1 min at 98 °C, (ii) 25 cycles of 45 s at 98 °C, 45 s at 60 °C,

and 30 s at 72 °C, and (iii) a final extension at 72 °C for 5 min. The PCR reaction mixture for archaeal DNA

amplification was similarly prepared but contained instead 1 µl of 25 mM $MgCl_2$ and 50 ng of isolated DNA. The

PCR protocol for archaeal DNA amplification included (i) initial denaturation for 1 min at 98 °C, (ii) 10 cycles of

45 s at 98 °C, 45 s at 63 °C, and 30 s at 72 °C, (iii) 15 cycles of 45 s at 98 °C, 45 s at 53 °C, and 30 s at 72 °C, and

(iv) a final extension at 72 °C for 5 min.

PCR products were checked by agarose gel electrophoresis and purified using the GeneRead Size Selection Kit

(QIAGEN GmbH, Hilden, Germany).

### 2.6.2. Data analysis and pipeline

Illumina PE sequencing of the amplicons and further process of the sequence data were performed in the Göttingen

Genomics Laboratory (Göttingen, Germany). After Illumina MiSeq processing, sequences were analyzed as

described in Egelkamp et al. (2017) with minor modifications. In brief, paired-end sequences were merged using

PEAR v0.9.10 (Zhang et al., 2014), sequences with an average quality score below 20 and containing unresolved

bases were removed with QIIME 1.9.1 (Caporaso et al., 2010). Non-clipped reverse and forward primer sequences

were removed by employing cutadapt 1.15 (Martin, 2011). USEARCH version 9.2.64 was used following the

UNOISE pipeline (Edgar, 2010). In detail, reads shorter than 380 bp were removed, dereplicated, and denoised

with the UNOISE2 algorithm of USEARCH resulting in amplicon sequence variants (ASVs) (Callahan et al.,

2017). Additionally, chimeric sequences were removed using UCHIME2 in reference mode against the SILVA

SSU database release 132 (Yilmaz et al., 2014). Merged paired-end reads were mapped to chimera-free ASVs and

an abundance table was created using USEARCH. Taxonomic classification of ASVs was performed with BLAST

against the SILVA database 132. Extrinsic domain ASVs, chloroplasts, and unclassified ASVs were removed from

the dataset. Sample comparisons were performed at same surveying effort, utilizing the lowest number of

sequences by random subsampling (20,290 reads for bacteria, 13,900 reads for archaea).

The paired-end reads of the 16S rRNA gene sequencing were deposited in the National Center for Biotechnology

Information (NCBI) in the Sequence Read Archive SRP156750.



## 3. Results

### 3.1. The Pompeia Province — geological settings

The Pompeia Province is situated in the Gulf of Cádiz offshore Morocco, within the so-called Middle Moroccan Field (Ivanov et al., 2000) at water-depths between 860 and 1000 m (**Fig. 1**). It compromises the active Al Gacel MV (**Fig. 1, C**), another mud volcano which is extinct (further referred as extinct MV) and two east-west elongated ridges (Northern Pompeia Coral Ridge and Southern Pompeia Coral Ridge). Scattered coral-mounds surround the ridges with a smooth relief (**Fig. 1, B**). CWCs were observed on seismic profiles resting on all these morphological features. Detailed geological profiles and 3D images of these features are shown in **Figs. 2** and **3**.

The Al Gacel MV is a cone-shape structure, 107 m high and 944 m wide, with its summit at 762 m depth and surrounded by a 11 m deep rimmed depression (León et al., 2012) (**Fig. 1, C**). It is directly adjacent to the Northern Pompeia Coral Ridge (**Fig. 2, A–B**), which extends ca. 4 km in westward direction (**Fig. 2, A–B**) and it is terminated by the Pompeia Escarpment (**Fig. 1, B; Fig. 2, C**). High resolution seismic profiles of the Pompeia Escarpment show CWC build-ups (R1 to R4) with steep lateral scarps of ca. 40 m height (**Fig. 2, C**). This MV is of sub-circular shape and exhibits a crater at its top (**Fig. 2, A–B**).

Ultra-high resolution sub-bottom seismic profile crossing the Pompeia Province from northwest (NW) to southeast (SE) (**Fig. 3, A**), shows (i) the Al Gacel MV surrounded by bottom-current deposits, (ii) an up to 130 m high CWC framework, growing on top the Southern Pompeia Coral Ridge, and (iii) semi-buried CWC mounds surrounding the ridge in areas of low relief. These CWC mounds locally form smooth, up to 25 – 30 m high top-rounded reliefs that are exposed, but then taper downward below the seafloor (applying sound speeds of 1750 m/s in recent sediments). Additionally, a multichannel seismic profile following the same track but with higher penetration below the seafloor (**Fig. 3, B**) shows high amplitude reflections inside the Al Gacel cone and enhanced reflections at the top of the diapirs (yellow dotted-line in **Fig. 3, B**), pointing to the occurrence of gas (hydrocarbon)-charged sediments. It furthermore exhibits breaks in seismic continuity and diapiric structures at different depths below the Southern Pompeia Coral Ridge and the Al Gacel MV, evidencing a fault system (**Fig. 3, B**). These tectonic structures may promote the development of overpressure areas (OP in **Fig. 3, B**) and consequent upward fluid flow to the surface.

### 3.2. ROV observation and measurements

Submersible ROV surveys at the Al Gacel MV (**Fig. 1, C**) revealed the presence of dispersed pockmark depressions at the eastern (Dive 10, 790 m) and northern flanks (Dive 11, 760 – 825 m depth). These sites are characterized by focused but low intensity seafloor bubbling (e.g. **Fig. 4, B**; **Fig. 5, A**). Analysis of water samples revealed $CH_4$-concentration up to 171 nM during Dive 10 and up to 192 nM during Dive 11 (Sánchez-Guillamón et al., 2015). Pockmarks were essentially formed by grey-olive mud breccia sediments and characterized by deposits of authigenic carbonates appearing in the center and edges, together with typical methane-seep related organisms (e.g. sulfide-oxidizing bacterial mats, chemosynthetic bivalves, siboglinid tubeworms) (**Fig. 4, B–C**; **Fig. 5**). Communities of non-chemosynthetic organisms (e.g. sponges, corals) were also found at pockmarks (**Fig. 4, B–C**; **Fig. 5, C**), but were more abundant in places where no seepage was detected (**Fig. 4, A**).

Observations with the submersible ROV at the Northern Pompeia Coral Ridge and the extinct MV (Dive 03) revealed widespread and abundant occurrences of dead scleractinian-corals (mainly *Madrepora oculata* and *Lophelia pertusa*) currently colonized by few non-chemosynthetic organisms (e.g. *Corallium tricolor*, other



octocorals, sea urchins) (**Fig. 6, B–D**). Locally, grey-black colored patches of sulfide-oxidizing bacterial mats
surrounded by dead chemosynthetic bivalves (*Lucinoma asapheus* and *Thysira vulcolutre*) were detected (**Fig. 6,**
**A**). $CH_4$-seepage appeared to be less than at the Al Gacel MV, with concentrations of 80 – 83 nM.
Water parameters display homogenous values between the four sampling sites (10 °C temperature, ca. 52 – 55 %
dissolved oxygen, ca. 31 Kg/m$^3$ density) (**Table 1**).

**3.3. Petrography and stable isotopes signatures of carbonates ($\delta^{18}O$, $\delta^{13}C$)**

Sample D10-R3 derives from a field of carbonates at the base of the Al Gacel MV which is inhabited by sponges
and corals (**Fig. 4, A**). The sample is a framestone composed of deep water scleractinian corals (*Madrepora* and
rare *Lophelia*) (**Fig. 7, A–B**). The corals are typically cemented by microbial automicrite (*sensu* Reitner et al.
1995) followed by multiple generations of aragonite. A matrix of dark allomicrite (*sensu* Reitner et al. 1995) with
oxidized framboidal pyrites and remains of planktonic foraminifera is restricted to few bioerosional cavities (ca.
5%) in the skeletons of dead corals (**Fig. 8, A–B**). $\delta^{13}C$ signatures of the matrix and cements range from −26.68 to
−18.38 ‰, while the embedded coral fragments exhibit $\delta^{13}C$ values between −5.58 and −2.09 ‰ (**Fig. 7, B; Table**
**2**). The $\delta^{18}O$ values generally range from +2.35 to +3.92 ‰ (**Fig. 9; Table 2**).
Sample D10-R7 was recovered from a pockmark on the eastern site of the Al Gacel MV that is virtually influenced
by active seepage (**Fig. 3, C**). It consists of black carbonate and exhibits a strong hydrogen sulfide ($H_2S$) odor (**Fig.**
**5, B**; **Fig. 7, C–D**). The top of this sample was inhabited by living octocorals (**Fig. 5, C**), while chemosymbiotic
siboglinid worms were present on the lower surface (**Fig. 5, D**). The sample is characterized by a grey peloidal
wackestone texture consisting of allomicrite with abundant planktonic foraminifers and few deep water miliolids.
The sample furthermore exhibits some fractured areas which are partly filled by granular and small fibrous cement,
probably consisting of Mg-calcite. Locally, light brownish crusts of microbial automicrite similar to ones in D10-
R3 are present (see above). Framboidal pyrite is abundant and often arranged in aggregates (**Fig. 8, C–D**). The
carbonate exhibits $\delta^{13}C$ values ranging from −28.77 to −21.13 ‰ and $\delta^{18}O$ values from +2.37 to +3.15 ‰ (**Fig. 9;**
**Table 2**).
Sample D11-R8 stems from an area with meter-sized carbonate blocks at the summit of the Al Gacel MV and is
mainly colonized by sponges and worms (**Fig. 4, D**). The sample generally exhibits a light grey mud- to wackestone
texture consisting of allomicrite with few scleractinian-coral fragments and planktonic foraminifers (**Fig. 7, E–F**).
The carbonate furthermore contains abundant quartz silt and, locally, pyrite enrichments. A further prominent
feature are voids that are encircled by dark grey halos and exhibit brownish margins (due to enrichments of very
small pyrite crystals and organic matter, respectively). $\delta^{13}C$ signatures of the matrix and cements range from
−14.82 to −14.74 ‰, while embedded coral fragments exhibit $\delta^{13}C$ values of −4.91 to −2.99 ‰ (**Fig. 7, F; Table**
**2**). $\delta^{18}O$ values generally range from +1.49 to +5.60 ‰ (**Fig. 9; Table 2**).
Sample D03-B1 is a necrotic fragment of a living scleractinian coral (*Madrepora oculata*) recovered from the
Northern Pompeia Coral Ridge (**Fig. 6, D**; **Fig. 7, G**). The coral-carbonate exhibits $\delta^{13}C$ values ranging from −8.08
to −1.39 ‰ and $\delta^{18}O$ values from −0.31 to +2.26 ‰ (**Fig. 9**; **Table 2**).

**3.4. Lipid biomarkers and compound specific carbon isotope signatures**

The hydrocarbon fractions of the sample D10-R7 mainly consist of the irregular, tail-to-tail linked acyclic
isoprenoids 2,6,11,15-tetramethylhexadecane ($C_{20}$; crocetane), 2,6,10,15,19-pentamethylicosane ($C_{25}$; PMI), as





well as of several unsaturated homologues of these compounds (**Fig. 10**). Additionally, it contains the regular,
head-to-tail linked acyclic isoprenoid pristane ($C_{19}$) and the cyclic isoprenoid hop-17(21)-ene.
The hydrocarbon fraction of sample D11-R8 is dominated by $n$-alkanes with chain-lengths ranging from $C_{14}$ to
$C_{28}$ (maxima at $n$-$C_{16}$ and, subordinated, at $n$-$C_{20}$ and $n$-$C_{28}$) (**Fig. 10**). The sample further contains pristane,
crocetane, the head-to-tail linked acyclic isoprenoid phytane ($C_{20}$) and traces of PMI.
Crocetane and PMI exhibited strongly depleted $\delta^{13}C$ values in sample D10-R7 (−101.2 ‰ and
−102.9 ‰, respectively), while they showed less depleted $\delta^{13}C$ values in sample D11-R8 (−57.2 ‰ and −74.3 ‰,
respectively). $\Delta^{13}C$ values of $n$-alkanes in sample D11-R8 ($n$-$C_{17-22}$) ranged between −30.8 ‰ and −33.0 ‰ (**Table**
**3**).

### 3.5. DNA inventories (MiSeq Illumina sequences)

Bacterial DNA (**Fig. 11, A**) from samples D10-R3 (authigenic carbonate, base of the Al Gacel MV) and D03-B1
(*Madrepora oculata* fragment, Northern Pompeia Coral Ridge) mainly derives from taxa that typically thrive in
the water-column (e. g. Actinobacteria, Acidobacteria, Chloroflexi, Bacteroidetes, Woeseiaceae, Dadabacteria,
Kaiserbacteria, Poribacteria, Planctomycetes, Gemmatimonadetes). The sample D10-R3 furthermore contains
bacterial DNA of the nitrite-oxidizing bacteria *Nitrospira sp.*, while the sample D03-B1 contains DNA of the
bacterial taxa Verrucomicrobia, Enterobacteria, *Nitrosococcus*. Noteworthy, one amplicon sequence variant
(ASV_189) with low number of clustered sequences has been found in D03-B1, identified as a methanotrophic
symbiont of *Bathymodiolus mauritanicus* (see Rodrigues et al., 2013).
Up to 50 % of bacterial DNA in sample D10-R7 (authigenic carbonate, top of the Al Gacel MV) derives from taxa
that are commonly associated with fluid seepage and AOM, i.e. sulfide-oxidizing bacteria, sulfate-reducing
bacteria (SRB) and methane-oxidizing bacteria. The most abundant are SRB taxa like SEEP-SRB1, SEEP-SRB2,
*Desulfatiglans*, *Desulfobulbus* and *Desulfococcus*, which typically form consortia with ANME archaea.
Archaeal DNA (**Fig. 11, B**) from samples D10-R3 and D03-B1 mainly consist of *Cenarchaeum sp.*, which
represents $70 - 90 \%$. *Candidatus Nitrosopumilus* is the second most abundant in both samples, representing 5 –
20 %. On the contrary, around 90 % of archaeal DNA in D10-R7 is related to ANME-1 and ANME-2 groups, in
good concordance with the relative abundances of SRB DNA.
Details of the number of reads per taxa are shown in the supplementary data, **Tables 1** and **2**.

### 4. Discussion

### 4.1. Evidence of hydrocarbon-rich seepage affecting the Pompeia Province

2D multichannel-seismic images show that the Pompeia Province is affected by fluid expulsion related to
compressional diapiric ridges and thrust faults (**Fig. 3, B**), as it has been reported from other areas of the Gulf of
Cádiz (Somoza et al., 2003; Van Rensbergen et al., 2005; Medialdea et al., 2009). There seem to be different types
of fault-conduit systems that link the overpressure zones (OP) with the seafloor (**Fig. 3, B**), controlling both type
and rate of seepage (e.g. eruptive, focused, diffused or dripping-like). At the Al Gacel MV, conduits are for
instance mainly linked to faults and a dense hydro-fracture network, allowing the migration of hydrocarbon-rich
muds from the overpressure zone to the surface. During active episodes, eruptions lead to the formation of mud-
breccia flows as observed in gravity cores (e.g. León et al., 2012). During rather dormant episodes, focused and
dripping-like seepage predominates, forming pockmark features (**Fig. 4, B**).





Currently, the Al Gacel MV is affected by continuous and focused dripping-like seepages. These sites of active
seepage are characterized by carbonates that are suspected to be methane-derived (e.g. sample D10-R7, **Fig. 4, B–**
**C**). In-situ ROV-measurements and subsequent water sample analysis demonstrated high proportions of $CH_4$ in
fluids that were escaping upon removal of the D10-R7 carbonate (171 nM; **Fig. 5, A**) (Sánchez-Guillamón et al.,
2015). This association suggests a genetic relationship between hydrocarbon-rich seepage and the carbonate, as
also evidenced by the low $\delta^{13}C$-values of the carbonates analyzed herein (down to ca. −30 ‰, **Fig. 9; Table 2**).
Indeed, the grey peloidal texture of this sample resembles that of AOM-derived automicrites from the Black Sea
that are related to micro-seepage of methane (cf. Reitner et al., 2005). The here observed isotopically depleted
acyclic isoprenoids such as crocetane and PMI ($\delta^{13}C$ values between ca. −103 and −57‰; **Fig. 10; Table 3**) are
typical fingerprints of AOM-associated Archaea (Hinrichs et al., 1999; Thiel et al., 1999, 2001; Peckmann et al.,
2001; Peckmann & Thiel, 2004), which is also in good accordance with the high abundance of DNA related to
ANME.  At the same time, abundant framboidal pyrite in the carbonate (**Fig. 8, C–D**) and SRB-related DNA (**Fig.**
**11**) evidences microbial sulfate reduction in the environment. All these evidences clearly demonstrate that the
carbonates have been formed via AOM, fueled by fluids from the underlying mud diapir.
Other carbonate samples from the Al Gacel MV (i.e. D10-R3 and D11-R8) probably have also been formed due
to AOM as they are also isotopically depleted ($\delta^{13}C$ values between ca. −25 and −15 ‰, **Fig. 9**, **Table 2**). However,
no active gas bubbling was observed during sampling, even though both samples still contain open voids which
could form pathways for a continuous migration of fluids.  In fact, several characteristics of these voids (e.g. dark
halos formed by pyrite, brownish margins due to organic matter enrichments) are very similar to those of methane-
derived carbonate conduits (cf. Reitner et al., 2015). This could imply that the intensity of hydrocarbon-rich
seepage and consequently AOM, may have fluctuated through time. The relatively low dominance of crocetane
and PMI in sample D11-R8 (**Fig. 10**), as well as their moderately depleted $\delta^{13}C$ values (−57.2 ‰ and −74.3 ‰,
respectively; **Table 3**), could be due to mixing effects and thus be in good accordance varying intensities of AOM
in the environment. Also, the presence of only few AOM-related DNA sequences (**Fig. 11**) and partly oxidized
pyrites in sample D10-R3 (**Fig. 8, A–B**) are well in line with this scenario. In concert it appears that the seepage
intensity has indeed been fluctuating.
There is no evidence for eruptive extrusions of muddy materials at the coral ridges. In the Southern Pompeia Coral
Ridge (**Fig. 3**), diapirs appears to rather promote an upward migration of hydrocarbon-rich fluids in a divergent
way throughout a more extensive seabed area. This results in a continuous and diffused seepage, which promotes
the occurrence of AOM and the formation of MDACs at the base of the ridges, related to the sulphate-methane
transition zone (SMTZ) (Boetius et al., 2000; Hinrichs and Boetius, 2002; González et al., 2012a). This is in good
accordance with the detection of methane (80 – 83 nM) at the Northern Pompeia Coral Ridge and the presence of
sulfide-oxidizing bacterial mats and shells of dead chemosynthetic bivalves at the western part of the ridge (**Fig.**
**6, A**). Likewise, the CWC Mounds Field surrounding the Southern Pompeia Coral Ridge (**Fig. 3**) is thoroughly
characterized by micro-seeps, due to ascending fluids from OPs through low-angel faults. This type of focused
seepage may promote formation of MDAC pavements in deeper layers of the sediments (**Fig. 3**), similar to coral
ridges along the Pen Duick Escarpment (Wehrmann et al., 2011). The generation of MDAC-hotspots at sites of
such seepage also explain the geometry of the downward tapering cones (**Fig. 3**).
**4.2. Ecological meaning of hydrocarbon-rich seepage for CWCs**



Our data suggests contemporaneous micro-seepage and CWC growth in the Pompeia Province (e.g. **Fig. 4, B**).
This relationship has also been observed elsewhere, e.g. in North Sea and off Mid Norway (Hovland, 1990;
Hovland & Thomsen, 1997), and the Angola margin (Le Guilloux et al., 2009). However, scleractinian fragments
recovered from the Al Gacel MV (embedded in carbonates D10-R3 and D11-R8) and the Northern Pompeia Coral
Ridge (D03-B1, necrotic part of a living *Madrepora oculata*) displayed barely depleted $\delta^{13}C$ values (ca. −8 to −1
‰; **Fig. 9**; **Table 2**), close to the $\delta^{13}C$ of marine seawater (0 ± 3 ‰, e.g. Hoefs, 2015). This does not support a
significant uptake of methane-derived carbon by the CWCs and thus a direct trophic dependency as previously
proposed (Hovland, 1990). Furthermore, the only DNA in sample D03-B1 that could be attributed to a potential
methanotrophic endosymbiont (ASV_189: Rodrigues et al., 2013) occurred in minor amounts and most likely
represents contamination from the environment or during sampling. Taken together, there is no evidence that
CWCs in the working area harbor microbial symbionts which potentially could utilize the hydrocarbon-rich fluids.
More likely, the CWCs feed on a mixture of phytoplankton, zooplankton and dissolved organic matter as
previously proposed for ones in other regions (Kiriakoulakis et al., 2005; Duineveld et al., 2007; Becker et al.,
2009; Liebetrau et al., 2010). This is in good accordance with the presence of DNA from various common archaeal
and bacterial taxa (e.g. Acidobacteria, Actinobacteria, Candidatus *Nitrosopumilus*, *Cenarchaeum sp.*) and some
potential members of the corals' holobiont (e.g. Enterobacteria, Verrucomicrobia, *Nitrosococcus sp.*) (Sorokin,
1995; Rädecker et al., 2015; Webster et al., 2016) in sample D03-B1 (**Fig. 11**).
CWC development and hydrocarbon-rich seepage are consequently linked *via* the formation of MDAC deposits,
which provide the hard substrata needed for CWC larval settlement (e.g. Díaz-del-Rio et al., 2003; Van Rooij et
al., 2011; Magalhães et al., 2012; Le Bris et al., 2016; Rueda et al., 2016). If too severe, however, fluid flow and
associated metabolic processes can result in local conditions that are lethal to CWCs (see 4.3). Moreover, AOM
fueled by fluid flow can also cause an entombment of the CWCs by MDACs (Wienberg et al., 2009, Wienberg &
Titschack, 2015), as observed in D10-R3 and D11-R8 carbonates from the Al Gacel MV (**Figs. 7 and 9**; **Tabs. 2**
**and 3**). It is therefore not surprising that large CWC systems in the Pompeia Province are always linked to
structures that are affected by rather mild, non-eruptive seepage (i.e. the extinct MV, the coral ridges and the CWC
Mound Fields: **Figs. 3 and 6**). The observation that these systems are in large parts "coral graveyards" (**Fig. 6, B–**
**D**), similar to other areas in the Gulf of Cádiz (see Foubert et al., 2008; Wienberg et al., 2009), may be explained
by a post-glacial decrease in current strength (Foubert et al., 2008). In the light of our findings, however, they
could also have been negatively affected by periods of intensive seepage during higher tectonic activity. Future
studies are important to test this hypothesis in greater detail.
**4.3. Spatio-temporal co-existence of CWCs and chemosynthetic organisms — the buffer effect**
As discussed above, MDAC deposits are ecologically beneficial for CWCs, as they served as optimal substrata
even when seepage is still present (e. g. Hovland, 1990; Hovland & Thomsen, 1997; Le Guilloux et al., 2009; this
study). Severe hydrocarbon-rich seepage, however, is ecologically stressful for the corals. Particularly, fluid- and
AOM-derived hydrogen sulfide is considered problematic because of its role in coral necrosis (Myers &
Richardson, 2009; García et al., 2016) and carbonate dissolution effects (Wehrmann et al., 2011).
Hydrogen sulfides can efficiently be buffered through the reaction with Fe-(oxyhydro)-oxides or $Fe^{2+}$ dissolved in
pore waters, ultimately forming pyrite (Wehrmann et al., 2011). Fe-(oxyhydro)-oxides nodules have previously
been observed in the Iberian and Moroccan margins (González et al., 2009; 2012b), but not in the Pompeia
Province. Instead, sulfide-oxidizing bacteria living in symbiosis with invertebrates (e.g. siboglinid worms:



Petersen & Dubilier, 2009) (**Fig. 5, D**) and thriving in mats (**Fig. 4, C**; **Fig. 6, A**) were particularly prominent along
this region. Furthermore, the consumption of methane and sulfate by AOM-microorganisms at active sites also
contribute to CWCs colonization of the carbonates by reducing environmental acidification.
An integrated model is proposed to represent the biological buffer effect observed in different cases along the
Pompeia Province. On the one hand, pockmark sites at the Al Gacel MV display the co-existence of non-
chemosynthetic corals (e.g. on top of D10-R7 carbonate; **Fig. 5**) with AOM-microorganisms and chemosynthetic
sulfide-oxidizing organisms (**Fig. 12, A**). Likewise, diapiric ridges (**Fig. 12, B**) and coral mounds (**Fig. 12, C**) may
similarly prevent CWCs dissolution, as observed in the Northern Pompeia Coral Ridge, where sulfide-oxidizing
bacterial mats were tightly related to the scleractinian-coral carbonates colonized by other non-chemosynthetic
octocorals (**Fig. 6**). This model represents the first approach on understanding the ecological linkage between
hydrocarbon-rich seepage and cold-water corals. The impact and exact capacity of this biological buffer, however,
remains elusive and must be evaluated in future studies.
**5. Conclusions**
The presence of cold-water corals related to hydrocarbon-seep structures like mud volcanoes and diapirs, is partly
due to the irregular topography affecting bottom water-currents, which supply nutrients to the corals. Likewise,
their tight-linkage to active hydrocarbon-rich seepage occurs by means of the production of methane-derived
carbonates and how they provide the hard substrata cold-water corals need to develop. The discovery of methane-
derived carbonates with embedded corals evidences the decline of coral colonization when the intensity of the
fluid seepage increases or becomes more violent. Consequently, cold-water coral growth in these habitats depends
directly on seepage intensity and how these fluids are drained onto the seafloor (i.e. eruptive, focused, diffused or
dripping-like). Furthermore, cold-water corals rely on the microbial AOM-metabolism and sulfide oxidation to
reduce seeped fluids in the environment, since they are harmful for the corals. This biological buffer is possibly
crucial to keep conditions favorable for the growth of cold-water corals in the studied area, particularly in times
of increased fluid seepage.
**Author contribution**
Blanca Rincón-Tomás, Dominik Schneider and Michael Hoppert carried out the microbial analysis. Jan-Peter
Duda carried out the biomarker analysis. Luis Somoza and Teresa Medialdea processed seismic and bathymetric
data. Pedro Madureira processed ROV data. Javier González and Joachim Reitner carried out the petrographic
analysis. Joachim Reitner carried out the stable isotopic analysis. Blanca Rincón-Tomás prepared the manuscript
with contributions from all co-authors.
**Competing interests**
The authors declare that they have no conflict of interest.
**Acknowledgments**
The authors thank the captain and the crew on board the R/V Sarmiento de Gamboa, as well as the UTM (Unidad
de Tecnología Marina), that have been essential for the success of this paper. Data obtained on board is collected



in the SUBVENT-2 cruise, which can be found in the IGME archive. This work was supported by the Spanish
project SUBVENT (CGL2012-39524-C02) and the project EXPLOSEA (CTM2016-75947) funded by the Spanish
Ministry of Science, Innovation and Universities.

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




**Table 1.** *In-situ* water variables measured during sampling with ROV sensors.

| | D10-R3 | D10-R7 | D11-R8 | D03-B1 |
|---|---|---|---|---|
| Temperature (℃) | 10.07 | 10.5 | 10.02 | 10.04 – 10.05 |
| Depth (m) | 850 – 890 | 791 | 763 | 829 |
| Conductivity (mS/cm) | 39.13 – 39.62 | 39.05 – 39.43 | - | - |
| Salinity (ppt) | - | - | 35.56 – 35.86 | 35.67 – 35.91 |
| Saturation of dissolved oxygen (%) | 53.64 – 54.69 | 54.02 – 54.35 | 51.95 – 53.92 | 52.46 – 56.22 |
| Dissolved oxygen (mg/l) | 4.81 – 4.90 | 4.85 – 4.88 | 4.66 – 4.84 | 4.71 – 5.09 |
| Density (kg/m$^3$) | 31.03 – 31.42 | 30.94 – 31.24 | 30.92 – 31.08 | 31.26 – 31.41 |




**Table 2.** Stable carbon and oxygen isotopes ($\delta^{13}$C, $\delta^{18}$O) of samples from the Al Gacel MV and the Northern
Pompeia Coral Ridge.

| Location | Sample | Identifier | $\delta^{18}$O (‰) | $\delta^{13}$C (‰) |
|---|---|---|---|---|
| Al Gacel MV | D10-R3 | 1 | 2.35 | −5.58 |
| | | 2 | 3.37 | −20.07 |
| | | 3 | 3.60 | −26.68 |
| | | 4 | 3.70 | −20.79 |
| | | 5 | 3.45 | −22.43 |
| | | 6 | 3.80 | −20.70 |
| | | 7 | 3.28 | −2.23 |
| | | 8 | 3.83 | −25.16 |
| | | 9 | 3.63 | −25.29 |
| | | 10 | 3.91 | −18.38 |
| | | 11 | 3.60 | −24.18 |
| | | 12 | 3.55 | −25.34 |
| | | 13 | 3.56 | −25.15 |
| | | 14 | 3.50 | −2.09 |
| | | 15 | 3.92 | −21.89 |
| | D10-R7 | 21 | 2.90 | −26.36 |
| | | 22 | 3.15 | −28.77 |
| | | 23 | 2.94 | −22.91 |
| | | 24 | 2.67 | −21.13 |
| | | 25 | 2.37 | −24.70 |
| | | 26 | 2.56 | −23.60 |
| | D11-R8 | 16 | 1.49 | −4.91 |
| | | 17 | 2.13 | −2.99 |
| | | 18 | 1.74 | −4.22 |
| | | 19 | 5.60 | −14.82 |
| | | 20 | 5.55 | −14.74 |
| Coral Ridge | D03-B1 | 1.1 | −0.38 | −7.93 |
| | | 1.2 | −0.86 | −7.77 |
| | | 1.3 | −0.51 | −7.35 |
| | | 1.5 | 1.15 | −5.26 |
| | | 1.4 | −1.03 | −8.08 |
| | | 1.6 | 0.69 | −5.96 |
| | | 1.7 | 0.54 | −6.42 |





**Table 2.** Continued

| Location | Sample | Identifier | $\delta^{18}O$ (‰) | $\delta^{13}C$ (‰) |
|----------|--------|-----------|---------------------|---------------------|
| Coral Ridge | D03-B1 | 3.1 | 1.59 | −2.08 |
| | | 3.2 | −0.31 | −6.27 |
| | | 3.3 | −0.89 | −6.78 |
| | | 3.4 | −0.94 | −6.73 |
| | | 3.5 | 1.84 | −2.21 |
| | | 3.6 | 2.26 | −1.39 |
| | | 3.7 | 1.74 | −2.87 |



**Table 3.** Stable carbon isotopic composition ($\delta^{13}C$) of selected lipid biomarkers (in **Figure 10**). (*) Please note
that crocetane in D11-R8 coelutes with phytane. n.d. = not detected.

| Compound | D10-R7 (‰) | D11-R8 (‰) |
|----------|------------|------------|
| $n$-C$_{17}$ | n.d. | −33.0 |
| $n$-C$_{18}$ | n.d. | −31.8 |
| $n$-C$_{19}$ | n.d. | −31.1 |
| $n$-C$_{20}$ | n.d. | −30.8 |
| $n$-C$_{21}$ | n.d. | −31.5 |
| $n$-C$_{22}$ | n.d. | −31.7 |
| Crocetane* | −101.2 | −57.2 |
| PMI | −102.9 | −74.3 |












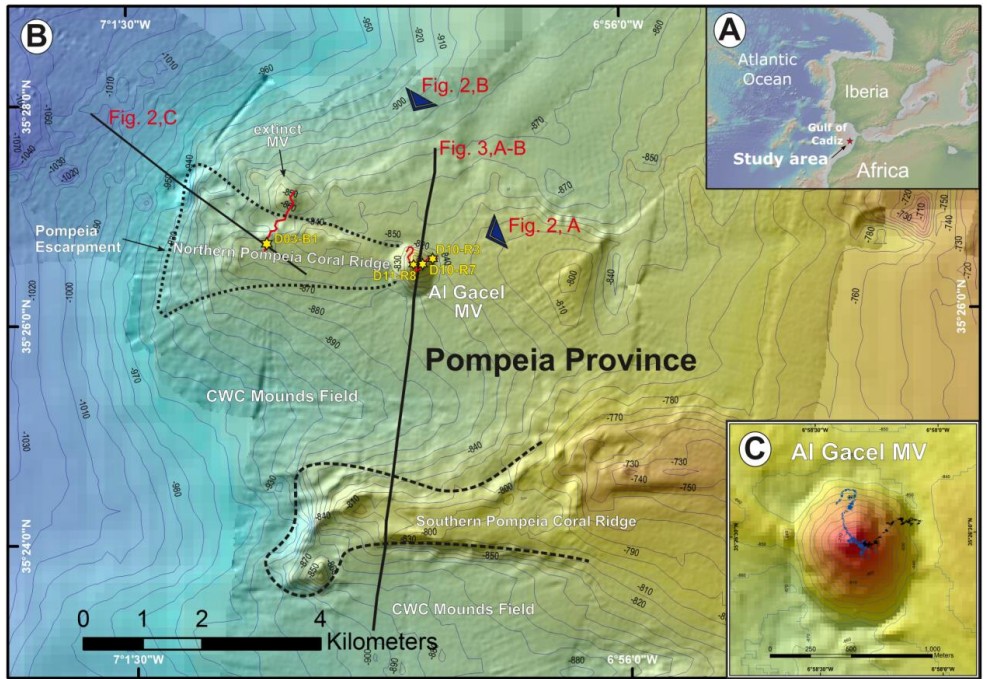


**Figure 1**. Bathymetric map of the study area. **A**: location of the Gulf of Cádiz between Spain, Portugal and Morocco. The study area is marked with a red star; **B**: the Pompeia Province including its different morphological features. Red lines indicate ROV-paths, yellow stars mark sampling sites; **C**: detailed map of the Al Gacel MV including pathways of Dive 10 and 11 (black and blue lines, respectively). Further details of the area are provided in **Figs. 2** and **3**.





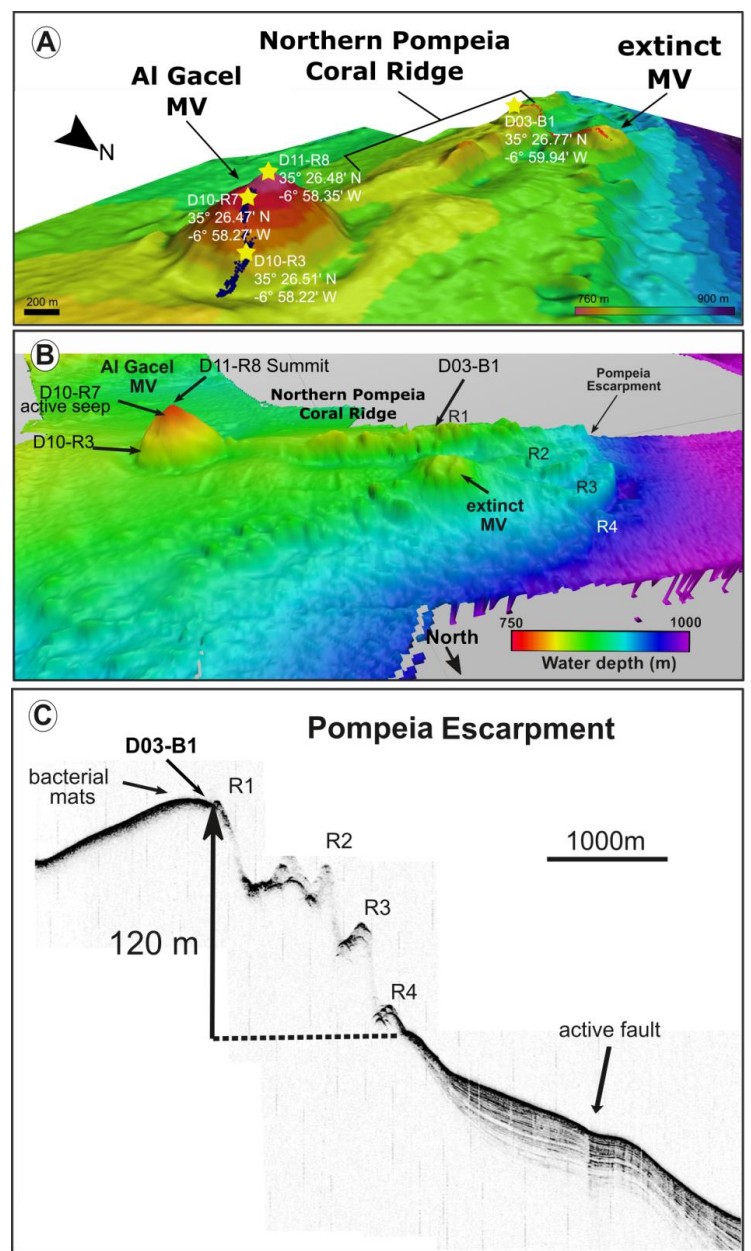

**Figure 2.** Bathymetric and seismic maps showing morphological features in the northern Pompeia Province. **A–**
**B**: bathymetric maps showing the Al Gacel MV, the Northern Pompeia Coral Ridge and the extinct MV. Yellow
stars mark sampling sites. **C**: seismic profile of the Pompeia Escarpment, westwards of the Northern Pompeia
Ridge.




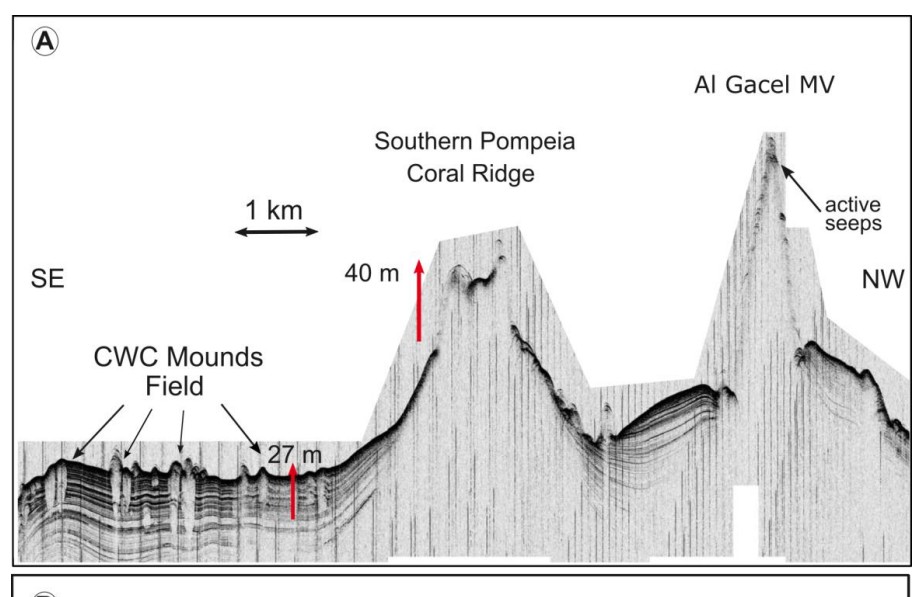

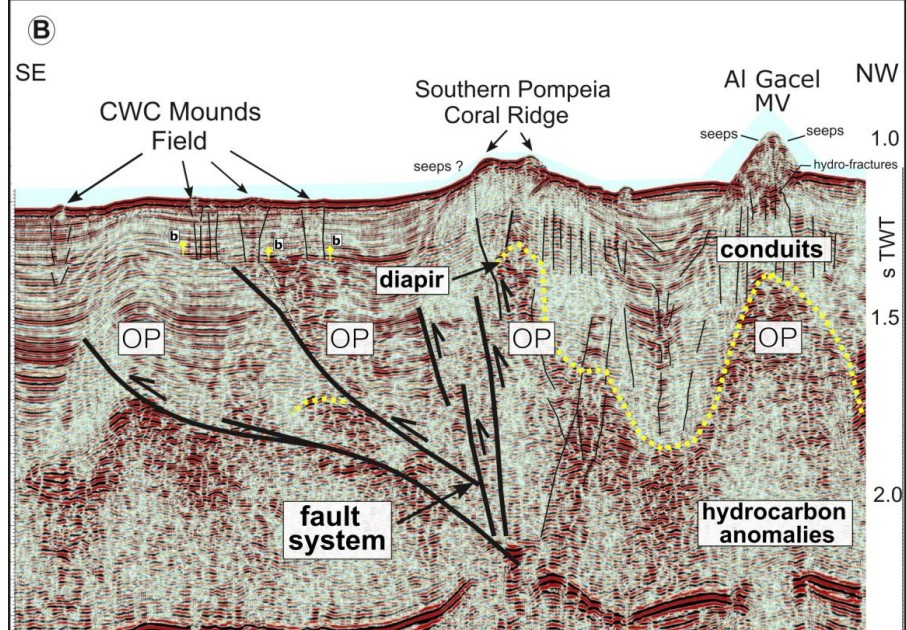


**Figure 3.** Seismic profiles showing geological features in the southern Pompeia Province. Note mud diapirism has
been described in this area (Vandorpe et al., 2017). OP = overpressure zone.

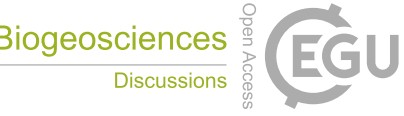



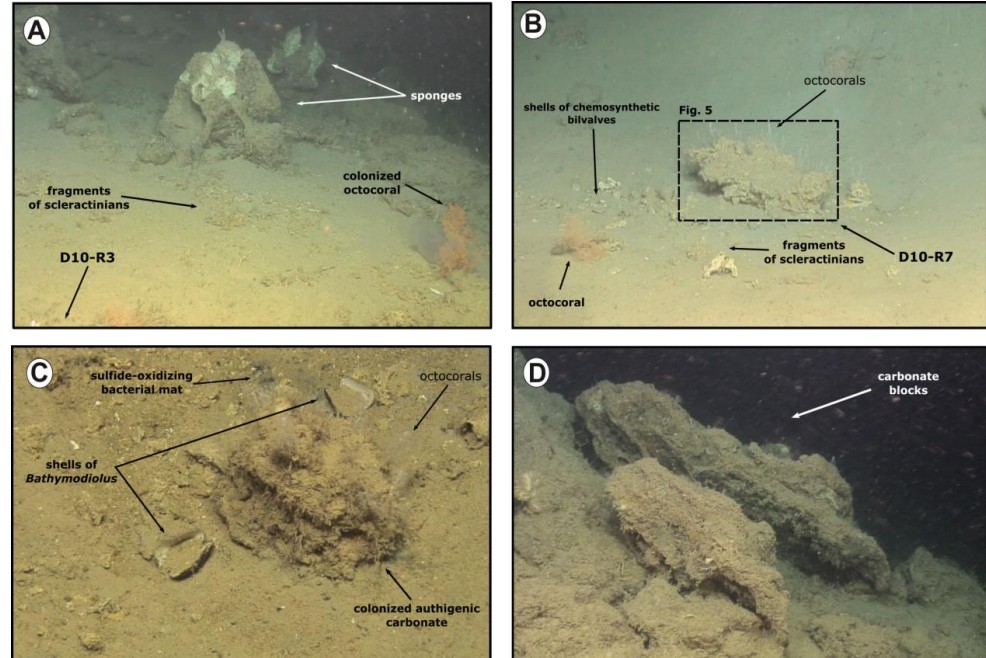

**Figure 4**: ROV still frames from the Al Gacel MV (Dives 10 and 11). **A**: eastern side of the volcano, displaying a
field of sponges, corals and carbonates; **B–C**: pockmark sites on the east side of the volcano, displaying authigenic
carbonate surrounded by shells of chemosynthetic bivalves, fragments of scleractinian and octocorals, as well as
sulfide-oxidizing bacterial mats; **D**: metric-sized carbonate blocks located in a slope at the summit of the volcano.





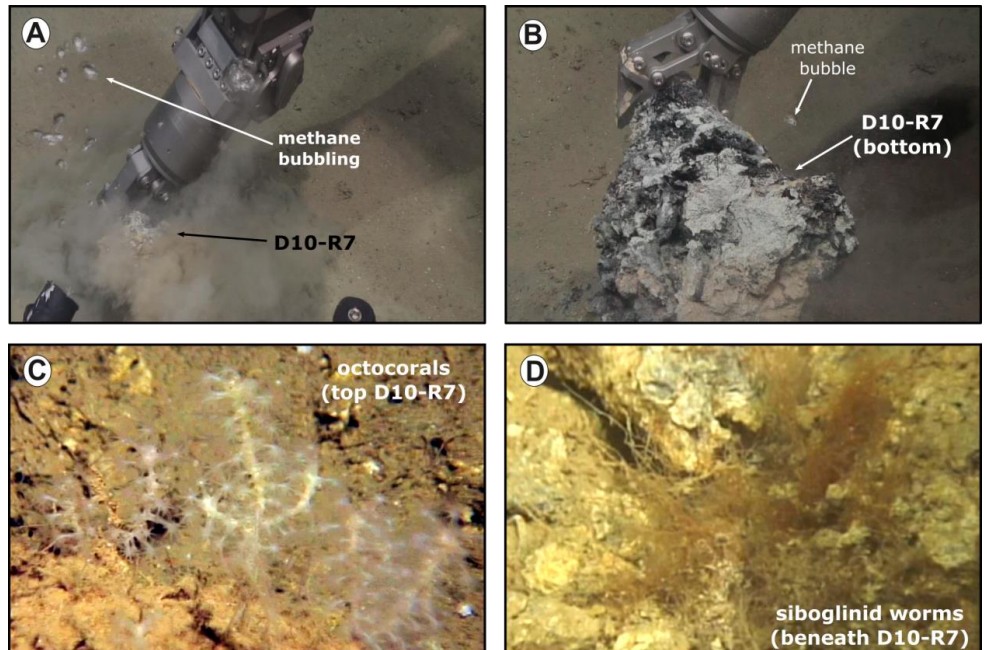


**Figure 5**: ROV still frames from the pockmark site shown in **Fig. 4**, **B**. **A–B**: release of bubbles while sampling;
**C**: detailed photograph of the octocorals on top of the carbonate; **D**: detailed still frame from siboglinid worms
beneath the carbonate.







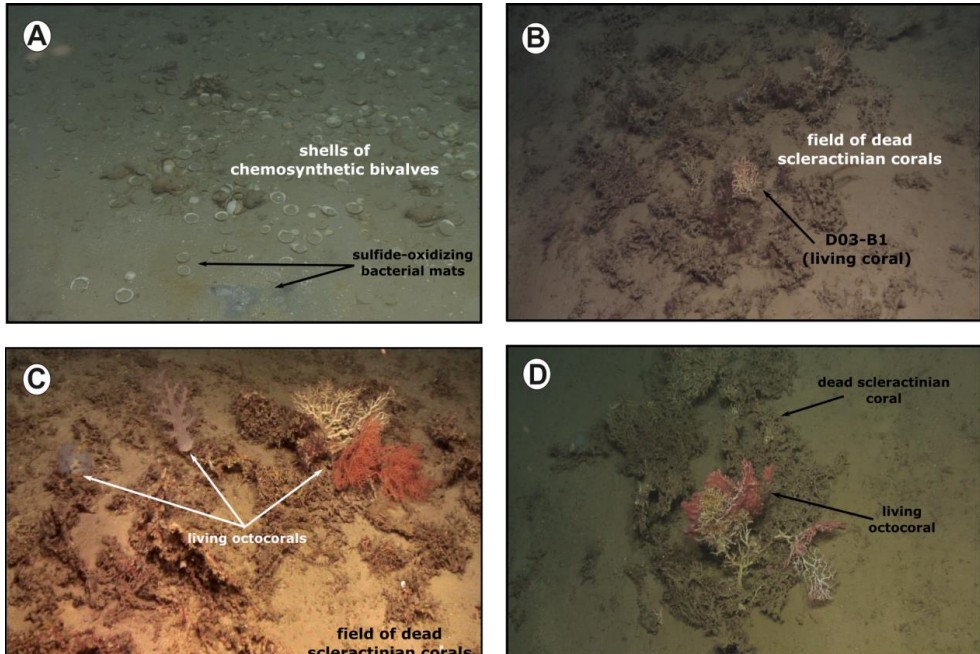


**Figure 6.** ROV still frames from the Northern Pompeia Coral Ridge and extinct MV (Dive 03). **A**: abundant shells

of chemosynthetic bivalves with sulfide-oxidizing bacterial mats at the western site of the Northern Pompeia Coral

Ridge; **B–D**: field of dead scleractinian-corals colonized by living corals; **D**: still frame from the extinct MV.




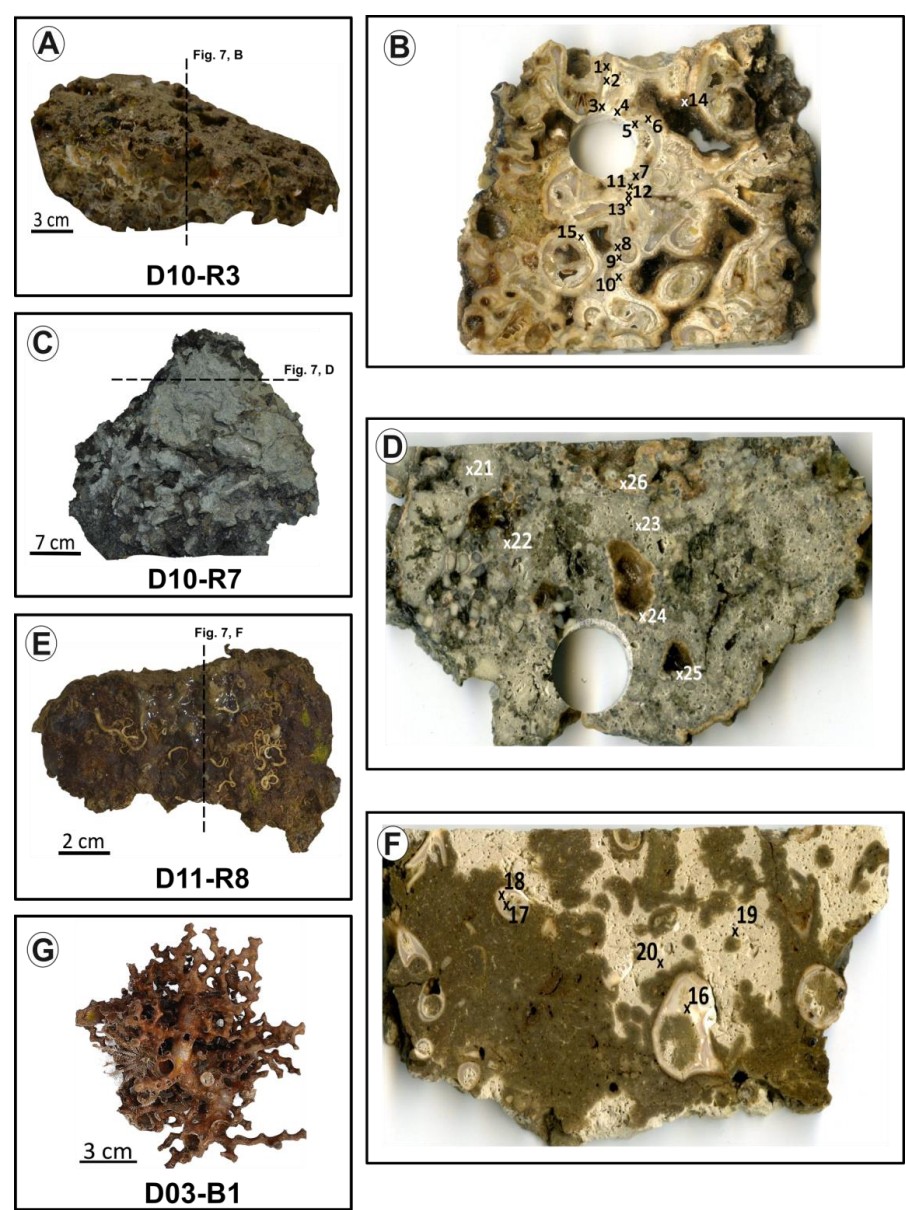


**Figure 7**. Photographs of analyzed samples including sampling sites for stable carbon and oxygen isotope ($\delta^{13}$C,
$\delta^{18}$O) analysis (crosses). **A–B**: D10-R3 carbonate with embedded corals; **C–D**: D10-R7 carbonate with strong H$_2$S
odor; **E–F**: D11-R8 carbonate with embedded corals; **G**: D03-B1 scleractinian-coral fragment, *Madrepora
oculata*.

778

779

780



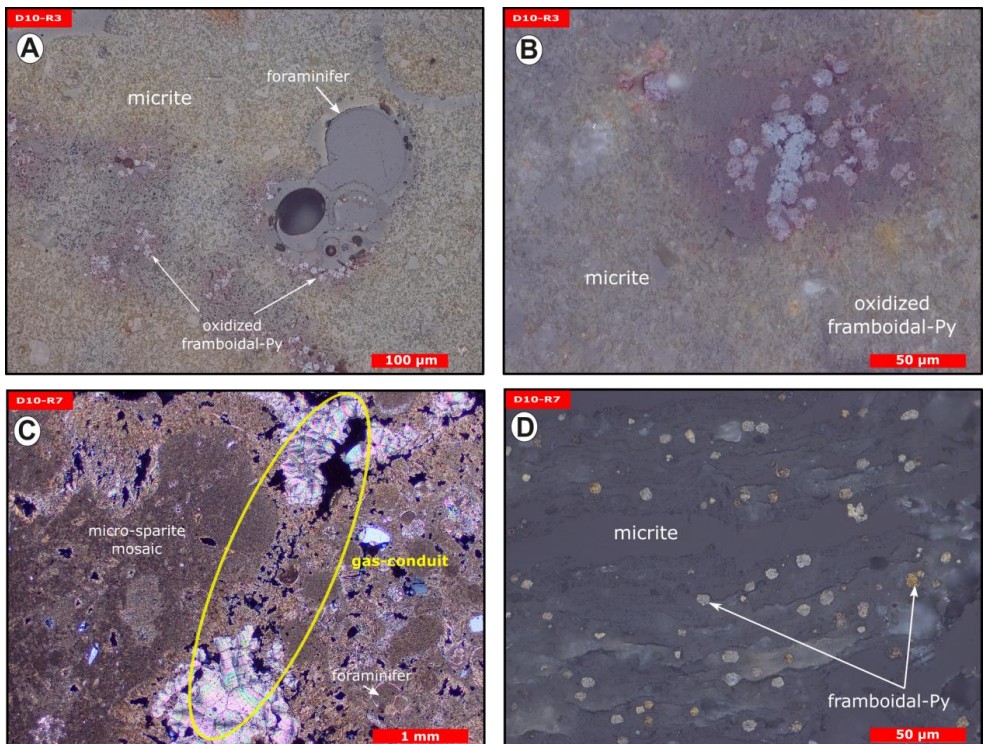

**Figure 8**. Thin section photographs of MDACs. **A–B**: D10-R3 consisting of a micritic matrix with scattered foraminifers and oxidized framboidal pyrites (reflected light); **C–D**: D10-R7 consisting of micritic and micro-sparitic carbonate with abundant unaltered framboidal pyrites (C, transmitted light; D, reflected light). Please note open voids which represent potential pathways for fluid seepage (yellow circle in C).



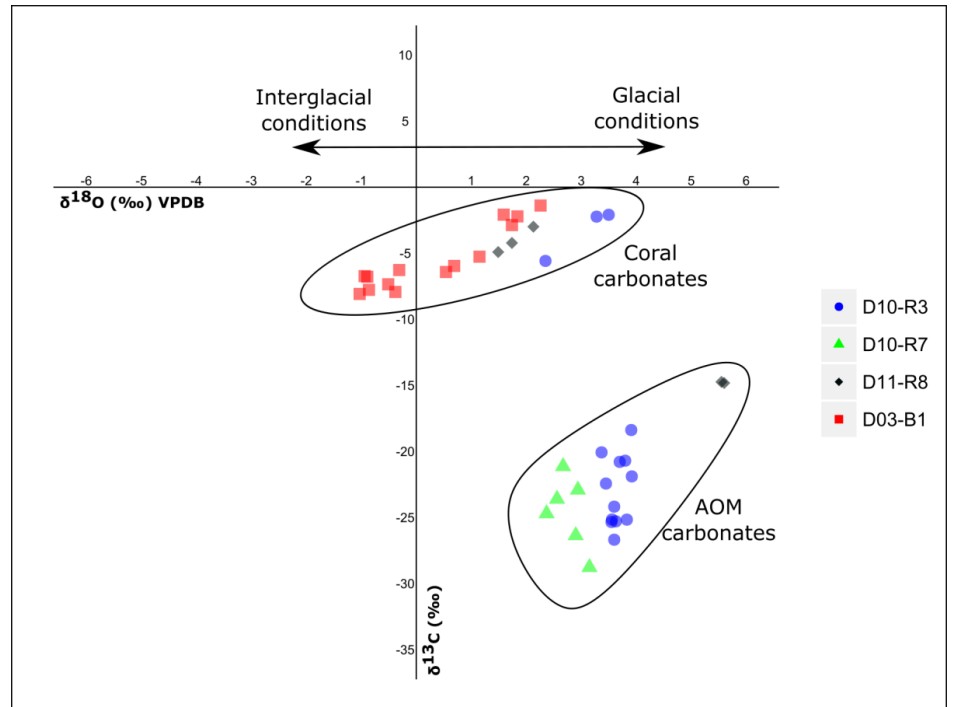

**Figure 9**. Stable carbon and oxygen isotopes ($\delta^{13}$C, $\delta^{18}$O) of samples from the Al Gacel MV and the Northern Pompeia Coral Ridge (see **Figure 3** for precise sampling points).

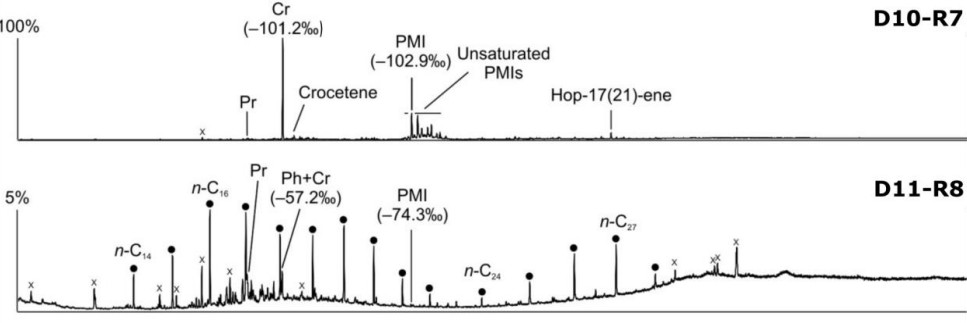

**Figure 10.** Total ion current (TIC) chromatograms of the analyzed samples. Isotopically depleted acyclic irregular isoprenoids such as Cr and PMI are typically found in settings influenced by the anaerobic oxidation of methane (AOM). Pr = pristane; Ph = phytane; Cr = crocetane; PMI = 2,6,10,15,19-pentamethylicosane; dots = n-alkanes; crosses = siloxanes (septum or column bleeding). Percentage values given on the vertical axes of chromatograms relate peak intensities to highest peak (Cr in D10-R7).



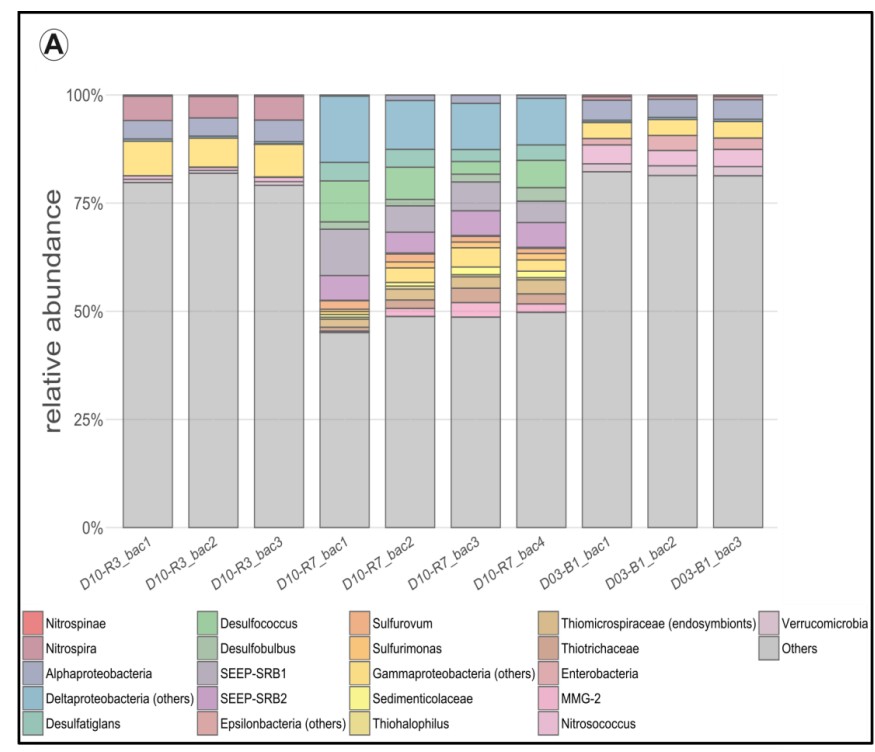

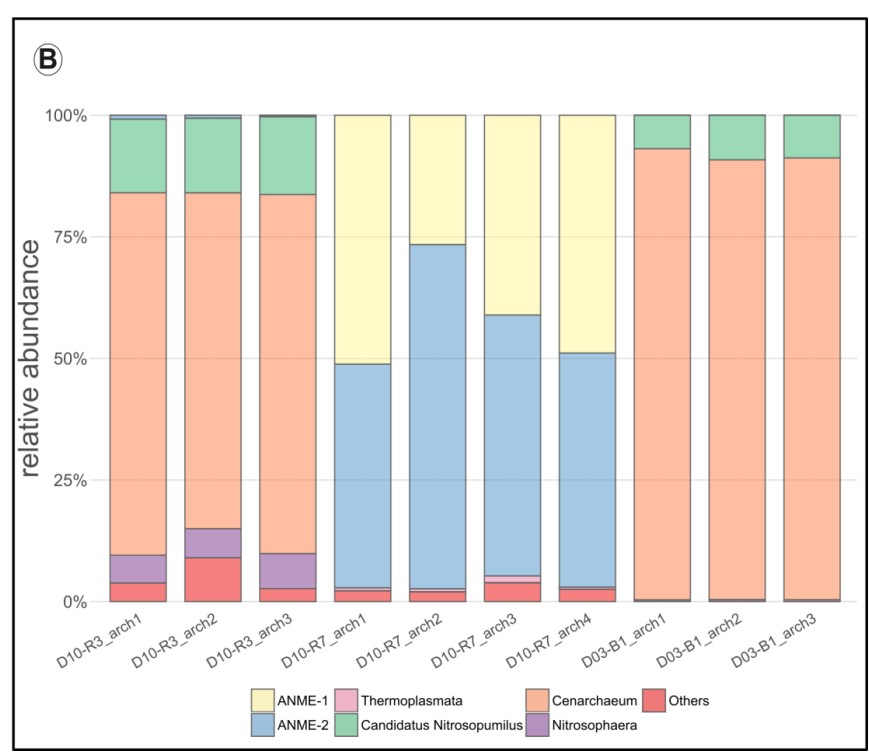





**Figure 11**. Bar chart representing the different taxa found in each sample according to relative abundances. A:
bacterial taxa; B: archaeal taxa. In "others" aggrupation is included taxa related to ubiquitous organism normally
found in sea- and seepage-related environments, and unclassified organisms. Number of reads per taxa detailed in
**Table S1** (bacteria) and **Table S2** (archaea).



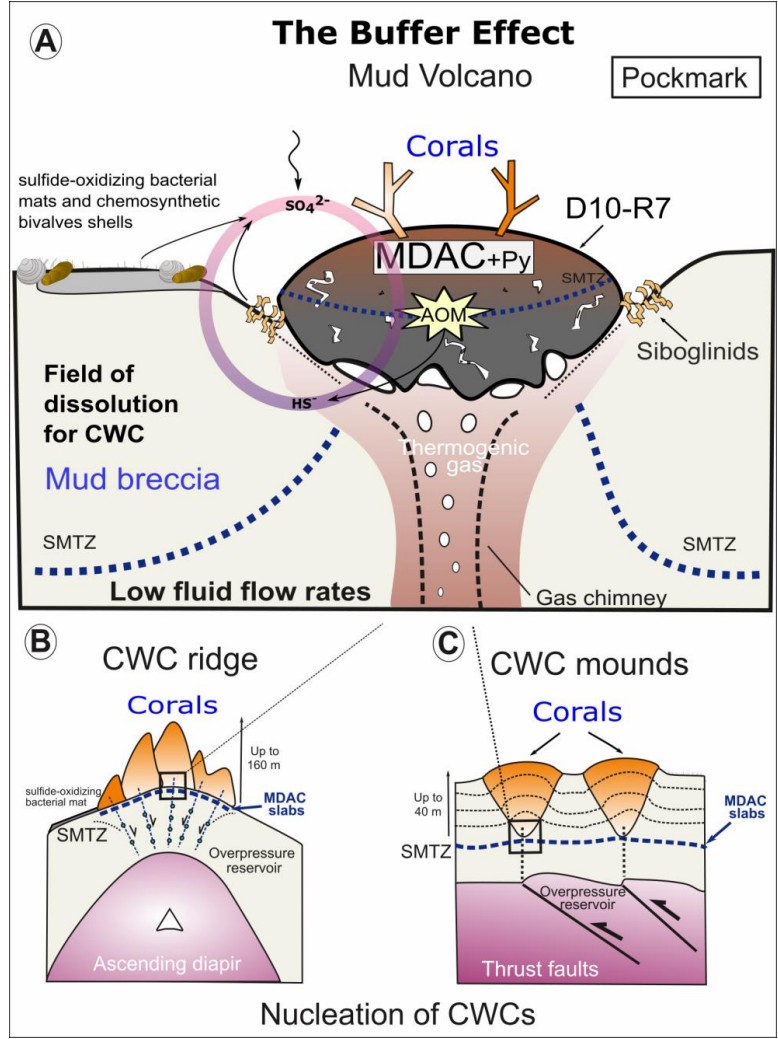

**Figure 12**. The buffer effect model. **A**: Buffer effect at pockmark sites (e.g. sampling site of D10-R7) where carbonates are formed directly on the bubbling site acting as a cap; **B**: Buffer effect at diapiric ridges where MDAC slabs are formed on the base of the ridge; **C**: Buffer effect at coral mounds where MDAC slabs are formed in deeper layers of the sediment. Py = pyrite, SMTZ: sulfur-methane transition zone.

810