# Peer review of "Cold-water corals and hydrocarbon-rich seepage in the 1 Pompeia Province (Gulf of Cádiz) — living on the edge 2"

_Biogeosciences, 2018_

## Referee Comment (RC1) · Anonymous Referee #1 · 3 Sep 2018

BG-2018-372

Title: Cold-water corals and hydrocarbon-rich seepage in the Pompeia Province (Gulf of Cádiz) - living on the edge

Author(s): Blanca Rincón-Tomás et al.

General Comments

The study examines relationships between cold water corals and fluid seepages from the sediment in a portion of the Gulf of Cadiz. The study addresses an interesting question and is fairly straightforward.

[Figure]

My concerns deal with the description of the experimental design. The four samples are mentioned. However, it is not clear to me how they were selected, i.e., whether by convenience, haphazardly, or with a design to test whether metabolism is fueled by fluids or sediment. Thus, it is not clear to me whether the data presented in the paper confirm the conclusions.

I am not saying that the study design was inappropriate. Rather it is not described well.

Also, the paper is a tough read, since much of the data are given as the author's collection codes rather than describing the sample. Thus, it is necessary to have two to three figures and the table placed side-by-side to understand what a value means.

Specific Comments

1) The abstract reads well.

2) The introduction reads well. One question is whether you have a testable hypothesis. Are you trying to ask whether the corals are fueled by fluids versus scavenging from currents. How are you going to distinguish between mechanisms?

3) In the methods please add section in which you describe the Experimental Design. How many samples were collected and from where? The descriptions of the laboratory methods are okay. However, I have no idea if you sampled thoroughly enough.

4) In Table 2, will readers know what Identifier means? I realize that the numbers correspond to pictures in the figures. However, it is very confusing to have to put the figure next to the table to interpret the data in the table. There must be a better way to present the data.

5) Rather than using code numbers for the sampling sites, it would help readers if you used descriptive names, such as 'active seep', etc.

6) Although amplicon sampling for microbial group is okay. Do you have evidence for microbial growth and activity? Perhaps in the discussion indicate which samples come

from fresh material and are likely to have fresh DNA versus samples in which the DNA could be old and preserved. I realized this is inferred by looking at the pictures, but again this is a convoluted way to present a story.

7) I suppose the model is okay. However, again a better presentation of the data might lead readers to the conclusion rather than relying on the author's story.

Technical Comments

1) Line 19: consider saying, 'rate a seepage via focused, scattered, diffused, etc.'

2) Line 34: change 'which' to 'that'.

3) Line 36: change to 'typically, they thrive, etc.'

4) Line 45: change 'ecological' to 'environmental' and 'are discussed to control' to 'influence'.

5) Line 51: delete 'e.g.'.

6) Line 53: change 'e.g.' to 'for example'.

7) Line 65: delete 'i.e.' and the parentheses. The text is not an example rather it is the description of 'coral graveyards.'

This is an okay study, and I suppose the conclusion is correct. However, the data presentation is convoluted. I do not have specific ways to present the data more clearly, but the author should be encouraged to try.

---

## Referee Comment (RC2) · Anonymous Referee #2 · 17 Sep 2018

This ms is a multidisciplinary investigation from a new research area of the links between cold-water corals and methane seepage, that adds considerable new data to the controversy, lending further support to the idea that cold-water corals are not nutritionally dependent on chemosynthetic methane or sulfur, but rather take advantage of methane derived carbonate deposits at seeps as hard substrate settlement. The ms finishes with a nice cartoon interpretation of the conditions of Gulf of Cadiz seeps and associated corals and seeps. The ms is well structured, explained and referenced, and has an adequate number of range of figures. Overall the data supports the interpretations, although I have one question about the inferences drawn from the delta C13 values of the coral skeletons (see below).
Detailed comments

Line 1. Title. The text after the hyphen: 'living on the edge' is unnecessary and adds nothing to the title. What edge? I suggest removing this.

Lines 26-27. Abstract Delta C13 values of the coral skeletons (see below)

Line 31. Abstract. Suggest 'seeping' rather than 'seeped' fluids.

Line 61. Suggest 'In addition' to replace 'On the other hand', as this is not a contrasting observation.

Line 76. 'Englobes' is not an English word. Seems like a transliteration of 'encompasses'.

Line 128. Don't start sentence with a number – spell it out.

Line 152. Can the authors give a little more detail of the nature of the samples used for the DNA work. Are these MDACs?

Lines 192-195. The background information about the Gulf of Cadiz isn't really results and would go better at the start of section 2.

Line 241 and other places. It's quite difficult at the moment to correlate the isotopic data in Table 2 with the sample points in Figure 7, because the specimen images in Figure 7 are not quite large enough to distinguish samples of authigenic carbonates from embedded coral skeletons. Therefore, could the authors add a column into Table 2 that makes it clear what the samples are for each of the isotopic data points, e.g. authigenic carbonate or coral skeleton.

Line 253. Replace 'stems' with 'comes'.

Line 254. In the figure the 'worms' look like serpulid worm tubes. Is this so? In which case please add this information.

Line 291. Replace 'On the contrary' with 'In contrast'.

[Figure]

Line 296. Spell out '2D' at start of sentence.

Line 305 and elsewhere. What is 'dripping-like' seepage? This isn't a description I recognize, so it would be helpful if the authors specify what this means.

Line 317. Suggest 'data', rather than 'evidences'.

Line 330. I'm unclear where is being referred to here.

Line 332. 'appear', not 'appears', as preceding diapirs is plural.

Line 339. Typo. Angle not angel.

Lines 346-354. The authors here suggest that the seawater-like values of the delta C13 from the dead scleractinian skeletons and those embedded in the MDAC show that the corals do not use methane as a food source, either directly or through symbionts. The authors need to be careful here, because some seep organisms that demonstrably do use methane (and sulfide) from seep fluids for food via endosymbionts produce carbonate skeletons that also have seawater-like delta C13 signatures. I am referring here to vesicomyid and bathymodiolin bivalves, that sequester seawater bi-carbonate ions to produce their shells. Using this model, having seawater-like delta C13 values in the coral skeletons does not prove that these animals do not use chemosynthetic food sources at the site. Really, to be able to settle this conclusively, authors would have to do isotopic, histological and DNA work on living corals from their site, not just on skeletal material and MDAC. In addition, it would be worth noting that scleractinian corals are found embedded in ancient seep carbonates too (see Goedert and Peckmann 2005); there may be some useful comparative isotopic data in that paper.

Lines 364-367. The entombment of coral skeletons by MDAC may have no consequence to corals, if they are already dead. It's not entirely clear from the text if the corals associated with the MDAC are dead or alive. If they are alive then this argument is stronger. Also, in most seep environments MDACs form in the subsurface where AOM reactions are occurring. Is this the case at this site? What proof is there of active

MDAC formation at the sediment-water interface, as indicated in Figure 12? This is pertinent to the arguments in section 4.3.

---

## Editor Decision (ED1)

**Start of review**

The manuscript "Cold-water corals and hydrocarbon-rich seepage in the Pompeia Province (Gulf of Cádiz) — living on the edge" by Rincón-Tomás et al. describes a study on the controls of CWC growth in an area subjected to hydrocarbon seepage. The manuscript is well written and contains interesting data. I do however find that there are some important issue that need to be addressed. The main points is that the rationale for the study design and sample analysis is not very clear until the Discussion.

Main issues:

• The authors write that the "This study aims at elucidating the linkage between the present-day formation of MDACs and CWCs development along the Pompeia Province (Fig. 1),", but it is not clear why the selected analysis is the best way to achieve this. For example, "Petrographic analysis" is described in the Methods but it is not clear why this analysis is necessary to answer the questions addressed in the manuscript.
• The suspected nutritional linkage between CWC and hydrocarbon seepage is known in the literature as the 'hydraulic theory' (see Hovland, Jensen et al. 2012 and references therein). The present study is a direct test of this theory in an area that is very suited to test this. The name "hydraulic theory" and/or related reference are however not mentioned in the manuscript (e.g. ln 50-52).

• Another major problem was description of the sampling design and the method of sampling. The authors write on line 84-86 "This study is based on collected data from the Pompeia Province, during the Subvent-2 cruise in 2014 aboard the R/V Sarmiento de Gamboa. The analysed samples were recovered from the Al Gacel MV (D10-R3, D10-R7, D11-R8) and the Northern Pompeia Coral Ridge (D03-B1) (Fig. 1)." This description is grossly inadequate. What was the sampling design? Are 'samples' collected ad random or based a preconceived plan? Why those sites? What material was sampled as 'the samples' (e.g. living coral pieces, coral rubble, sediment with rubble, carbonates)? Size/weight of the samples? Number of samples? Replication? How are the samples taken (ROV arm, push core)? How were samples stored on the ROV, how long before samples reached the surface how are samples processed/stored on-board (significant given the DNA/RNA analysis, e.g. with respect to cross contamination, microbial community shifts)?

• The authors are addressing ecological questions (see e.g. line 34-38, line 50-52 and line 75 "...present-day formation of MDACs and CWCs development...") using studies of carbonates. One of the issues that is particularly relevant for the interpretation of these data is whether the analysis was performed on carbonates with living CWC or not. From the pictures and description it seems plausible that only dead CWC carbonates were studied (although ln 348 mentions "the necrotic part of living Madrepora"), but this begs the question how representative the RNA/DNA/biomarker analysis is when only carbonates of dead CWCs are studied. To what extent do the authors think that the organic components of the carbonates still represent the CWC microbial community?

Similarly for the 13C carbonate analysis, is it known well enough whether CWCs leave a distinct isotope mark in the carbonates that is representative for feeding on surface-derived organic matter versus hydrocarbons? Targeted sampling of also living CWC pieces and comparison with the sampled carbonates would have provided a means to address this.

• The authors mention that the ROV had sensors for CO2 and CH4 data and could take NISKIN water samples for CH4. In the results section (ln 219-221 and ln 231) CH4 data are mentioned but in the M&M nothing can be found on sampling location (e.g. height above sediment), sensor calibration, samples handling, sample analyses of the water samples.

• The site description in 3.1 should be partly moved to the Materials and Methods. Only the new results from this study should stay in 3.1.

• The authors infer that "severe seepage results in lethal conditions for CWCs" (line 363 - 364 and 377-378), but I see no evidence for that in the paper. In addition, the authors concluded that CWCs can be entombed by MDAC formation, it is however not clear whether this entombment is the cause of CWC mortality or that this entombment took place after CWC demise following for example from post-glacial decrease in current strength.

Suggestions for minor edits:
ln 48-50: reduce number of refs
ln 59: reduce number of refs
ln 72-73: reduce number of refs

ln 112: Please also give the values of the VPDB used, to avoid confusion
ln 124: "have a global distribution" in stead of "globally widespread"
ln 152: replace "... solid samples were…" with "…sample material was…"
ln 230: replace "…by dead.." with "… by shells of the chemosynthetic bivalves Lucinoma…"
ln 243: What does "virtually influenced" mean?
ln 262: "… values ranging from…". From the methods it is unclear on what this range is based, replication, multiple samples?
ln 307: What does "proportions" here mean? Do you mean "rates" or "concentrations"?
ln 308: So was methane sampled upon removal of the carbonate blocks?
ln 368: The authors also mentioned the availability of a CO2 sensor on the ROV. Has this been used to measure aragonite saturation states at the different locations?
ln 755: Fig 4C. There is a black pointing to "octocorals", but I cannot see these on the picture.

Bibliography
Hovland, M., S. Jensen and T. Indreiten (2012). "Unit pockmarks associated

withLopheliacoral reefs off mid-Norway: more evidence of control by 'fertilizing' bottom currents." Geo-Marine Letters 32(5-6): 545-554.

**END OF REVIEW**

---

## Author Response (AR2)

**Authors response to Referee n° 1**

We are thankful for the constructive and helpful comments that have helped us to improve our manuscript. We are aware that the manuscript holds a high amount of data which can be difficult to follow at some points and tried to keep it as concise as possible. We considered all comments carefully and modified and followed most of the suggestions.

*Specific Comments from Referee n° 1*

**2) The introduction reads well. One question is whether you have a testable hypothesis. Are you trying to ask whether the corals are fueled by fluids versus scavenging from currents. How are you going to distinguish between mechanisms?**

**Response**: the aim of the study is to address the linkage between CWCs and present day formation of MDACs in the Pompeia Province. For this purpose, we combined analyses of ROV images, geophysical data and sample materials. For instance, we analyzed $\delta^{13}C$ signatures of coral skeletons to evaluate whether these organisms were directly relying on $CH_4$. We found that the coral skeletons exhibited significantly higher $\delta^{13}C$ values than the co-occurring AOM-derived carbonates, thus not supporting $CH_4$ as important carbon source. Rather, the corals were feeding on material suspended in currents.

**3) In the methods please add section in which you describe the Experimental Design. How many samples were collected and from where? The descriptions of the laboratory methods are okay. However, I have no idea if you sampled thoroughly enough.**

**Response:** we included more detailed information on our sample strategy and study design in the material and methods section.

**4) In Table 2, will readers know what Identifier means? I realize that the numbers correspond to pictures in the figures. However, it is very confusing to have to put the figure next to the table to interpret the data in the table. There must be a better way to present the data.**

**Response**: done. We replaced "Identifier" by "Identification number in Fig. 7". In Addition, we added an additional column to the table in which we provide information on the analyzed material.

**5) Rather than using code numbers for the sampling sites, it would help readers if you used descriptive names, such as 'active seep', etc.**

**Response:** done. We have revised the use of code numbers throughout the manuscript.

**6) Although amplicon sampling for microbial group is okay. Do you have evidence for microbial growth and activity? Perhaps in the discussion indicate which samples come from fresh material and are likely to have fresh DNA versus samples in which the DNA could be old and preserved. I realized this is inferred by looking at the pictures, but again this is a convoluted way to present a story.**

**Response**: we have improved the information concerning the DNA material related to each sample in the manuscript, and we have specified the type of sample from which the DNA has been extracted (lines 183–186 in the revised manuscript). Furthermore, we added some extra information in Fig. 11 to clarify and remain the type of sample. DNA analyses cannot conclude if DNA is "old" or "fresh", but we can estimate (together with other analyses) if the sample used for this analysis is fresh or not. but we can infer this by assessing the relative age and preservation of the analyzed sample. For instance, an AOM-derived carbonate recovered from an active pockmark (sample D10-R7) exhibits more DNA of AOM-related microorganisms (ANME and SRB) than oxidized AOM-derived carbonates recovered from regions that are currently not affected by seepage (sample D10-R3).

**7) I suppose the model is okay. However, again a better presentation of the data might lead readers to the conclusion rather than relying on the author's story.**
**Response**: done. We have modified the last paragraph of the section 4.3. for a better understanding of our model (lines 439–446 in the revised manuscript).

*Technical Comments from Referee n° 1*
**1) Line 19: consider saying, 'rate a seepage via focused, scattered, diffused, etc.'**
**Response**: done. We revised the sentence to "the type of seepage such as focused, scattered, diffused or eruptive".

**2) Line 34: change 'which' to 'that'.**
**Response**: done.

**3) Line 36: change to 'typically, they thrive, etc.'**
**Response**: done.

**4) Line 45: change 'ecological' to 'environmental' and 'are discussed to control' to 'influence'.**
**Response**: done.

**5) Line 51: delete 'e.g.'.**
**Response**: done.

**6) Line 53: change 'e.g.' to 'for example'.**
**Response**: done.

**7) Line 65: delete 'i.e.' and the parentheses. The text is not an example rather it is the description of 'coral graveyards.'**
**Response**: done.

**Authors response to Referee n° 2**

We are thankful for your constructive feedback and the helpful comments. We have considered and addressed your suggestions carefully, and almost all have been followed in the revised manuscript.

*Detail Comments from Referee n° 2*

**1) Line 1. Title. The text after the hyphen: 'living on the edge' is unnecessary and adds nothing to the title. What edge? I suggest removing this.**

**Response:** we would like to keep the text "living on the edge" to emphasize that hydrocarbon-rich seepage has both advantages and disadvantages for cold-water corals growth.

**2) Lines 26-27. Abstract Delta C13 values of the coral skeletons (see below)**

**Response:** see discussion on reviewer comment n° 19 below.

**3) Line 31. Abstract. Suggest 'seeping' rather than 'seeped' fluids.**

**Response**: done.

**4) Line 61. Suggest 'In addition' to replace 'On the other hand', as this is not a contrasting observation.**

**Response:** done.

**5) Line 76. 'Englobes' is not an English word. Seems like a transliteration of 'encompasses'.**

**Response**: done.

**6) Line 128. Don't start sentence with a number – spell it out.**

**Response**: done.

**7) Line 152. Can the authors give a little more detail of the nature of the samples used for the DNA work. Are these MDACs?**

**Response**: done. We now provide more information on the nature of the samples (lines 182–185 in the revised manuscript).

**8)Lines 192-195. The background information about the Gulf of Cadiz isn't really results and would go better at the start of section 2.**

**Response**: we agree that the background information of the Gulf of Cádiz is not part of results. However, the Pompeia Province region, which our study is focused on, has not been described in detail so far. We here provide the first description of geological structures in this area (Southern and Northern Pompeia Coral ridges, Cold-water Coral Mounds Fields), including novel data (e.g., bathymetry, seismics). For this reason, we consider it appropriate to report these findings in the results sections.

**9) Line 241 and other places. It's quite difficult at the moment to correlate the isotopic**

data in Table 2 with the sample points in Figure 7, because the specimen images in Figure 7 are not quite large enough to distinguish samples of authigenic carbonates from embedded coral skeletons. Therefore, could the authors add a column into Table 2 that makes it clear what the samples are for each of the isotopic data points, e.g. authigenic carbonate or coral skeleton.

**Response:** done. One more column has been added in Table 2 as proposed, indicating the type of samples from which stable isotopic analyses are.

**10) Line 253. Replace 'stems' with 'comes'.**

**Response**: done.

**11) Line 254. In the figure the 'worms' look like serpulid worm tubes. Is this so? In which case please add this information.**

**Response**: done.

**12) Line 291. Replace 'On the contrary' with 'In contrast'.**

**Response:** done.

**13) Line 296. Spell out '2D' at start of sentence.**

**Response:** done.

**14) Line 305 and elsewhere. What is 'dripping-like' seepage? This isn't a description I recognize, so it would be helpful if the authors specify what this means.**

**Response**: done. "Dripping-like refers to intermittent bubbling fluids" (lines 343–344 in the revised manuscript).

**15) Line 317. Suggest 'data', rather than 'evidences'.**

**Response**: done.

**16) Line 330. I'm unclear where is being referred to here.**

**Response**: removed.

**17) Line 332. 'appear', not 'appears', as preceding diapirs is plural.**

**Response:** done.

**18) Line 339. Typo. Angle not angel.**

**Response**: done.

**19) Lines 346-354. The authors here suggest that the seawater-like values of the delta C13 from the dead scleractinian skeletons and those embedded in the MDAC show that the corals do not use**

methane as a food source, either directly or through symbionts. The authors need to be careful here, because some seep organisms that demonstrably do use methane (and sulfide) from seep fluids for food via endosymbionts produce carbonate skeletons that also have seawater-like delta C13 signatures. I am referring here to vesicomyid and bathymodiolin bivalves, that sequester seawater bi-carbonate ions to produce their shells. Using this model, having seawater-like delta C13 values in the coral skeletons does not prove that these animals do not use chemosynthetic food sources at the site. Really, to be able to settle this conclusively, authors would have to do isotopic, histological and DNA work on living corals from their site, not just on skeletal material and MDAC. In addition, it would be worth noting that scleractinian corals are found embedded in ancient seep carbonates too (see Goedert and Peckmann 2005); there may be some useful comparative isotopic data in that paper.

**Response:** We included the paper by Goedert and Peckmann, 2005. We fully agree that analyses of coral tissues ($\delta^{13}C$, DNA) would add important information on their nutrition and metabolic relationships. However, we still regard $\delta^{13}C$ values of their skeletons as valuable proxy for the possible uptake of $CH_4$. Corals utilize $HCO_3^-$ deriving from both the environment and the internal production of $CO_2$ for skeleton biomineralization (Swart, 1983; Zoccola et al., 2015; Nakamura et al., 2018). Therefore, if they uptake $CH_4$ as a carbon source, the $CO_2$ produced from $CH_4$ metabolism would be used, and consequently parts of the $HCO_3^-$ utilized for biomineralization would be isotopically depleted. This "mixing effect" would result in at least partially depleted $\delta^{13}C$ values of the skeletons, similar to some chemosynthetic vesicomyid and lucinid bivalves (Hein et al., 2006). The skeletons of the corals analyzed herein, however, exhibit significantly higher $\delta^{13}C$ values than the co-occurring AOM-derived carbonates. Thus, they are not indicative for $CH_4$ as important carbon source.

**20) Lines 364-367. The entombment of coral skeletons by MDAC may have no consequence to corals, if they are already dead. It's not entirely clear from the text if the corals associated with the MDAC are dead or alive. If they are alive then this argument is stronger. Also, in most seep environments MDACs form in the subsurface where AOM reactions are occurring. Is this the case at this site? What proof is there of active MDAC formation at the sediment-water interface, as indicated in Figure 12? This is pertinent to the arguments in section 4.3.**

**Response:** We cannot determine if the scleractinian corals embedded in AOM-derived carbonates (samples D10-R3 and D11-R8) were alive or dead when they were buried (lines 812-813 in the revised manuscript). However, we observed living corals in areas that are currently affected by seepage (e.g. the Northern Pompeia Coral Ridge, lines 262–263 in the revised manuscript; Fig. 6, C). Furthermore, we observed living octocorals growing on surfaces of currently formed AOM-derived carbonates (e.g., in an active pockmark in the Al Gacel MV, sample D10-R7; Fig. 5, C). These observations imply that corals in these regions are directly affected by methane seepage and the microbially mediated formation of carbonates due to AOM.

*References*

Hein, J. R., Normark, W. R., McIntyre, B. R., Lorenson, T. D., and Powell, C. L.: Methanogenic calcite, 13C-depleted bivalve shells, and gas hydrate from a mud volcano offshore southern California, Geology, 34(2), 109–112, 2006.

Nakamura, T., Nadaoka, K., Watanabe, A., Yamamoto, T., Miyajima, T., and Blanco, A. C.: Reef-scale modeling of coral calcification responses to ocean acidification and sea-level rise, Coral Reefs, 37, 2018.

Swart, P. K.: Carbon and Oxygen Isotope Fractionation in ScleracUnian Corals: a Review, Earth-Sci. Rev., 19, 51–80, 1983.

Zoccola, D. Ganot, P., Bertucci, A., Caminit-Segonds, N., Techer, N., Voolstra, C. R., Aranda, M., Tambutté, E., Allemand, D., Casey, J. R., and Tambutté, S.: Bicarbonate transporters in corals point towards a key step in the evolution of cnidarian calcification, Sci. rep.-UK, 5, 2015.

**Authors response to Referee n° 3**

We are thankful for your useful and interesting comments. We hope we have addressed successfully the different issues discussed here.

*Main issues*

-**The authors write that the "This study aims at elucidating the linkage between the present-day formation of MDACs and CWCs development along the Pompeia Province (Fig. 1),", but it is not clear why the selected analysis is the best way to achieve this. For example, "Petrographic analysis" is described in the Methods but it is not clear why this analysis is necessary to answer the questions addressed in the manuscript. The suspected nutritional linkage between CWC and hydrocarbon seepage is known in the literature as the 'hydraulic theory' (see Hovland, Jensen et al. 2012 and references therein). The present study is a direct test of this theory in an area that is very suited to test this. The name "hydraulic theory" and/or related reference are however not mentioned in the manuscript (e.g. ln 50-52)**

**Response**: The "hydraulic theory" is now included in the introduction with references (line 55 of revised manuscript). Petrographic analyses are needed to be sure that these are seep carbonates, and to find the right sampling points for isotope analysis — we have to discriminate between authigenic carbonates, corals, micritic phases. of samples. For instance, embedded corals in some of the AOM-carbonates (D10-R3 and D11-R8) have been described and discriminated from the AOM-carbonate facies by petrographic analysis.

-**Another major problem was description of the sampling design and the method of sampling. The authors write on line 84-86 "This study is based on collected data from the Pompeia Province, during the Subvent-2 cruise in 2014 aboard the R/V Sarmiento de Gamboa. The analysed samples were recovered from the Al Gacel MV (D10-R3, D10-R7, D11-R8) and the Northern Pompeia Coral Ridge (D03-B1) (Fig. 1)." This description is grossly inadequate. What was the sampling design? Are 'samples' collected ad random or based a preconceived plan? Why those sites? What material was sampled as 'the samples' (e.g. living coral pieces, coral rubble, sediment with rubble, carbonates)? Size/weight of the samples? Number of samples? Replication? How are the samples taken (ROV arm, push core)? How were samples stored on the ROV, how long before samples reached the surface how are samples processed/stored on-board (significant given the DNA/RNA analysis, e.g. with respect to cross contamination, microbial community shifts)?**

**Detailed response to "***What was the sampling design? Are 'samples' collected ad random or based a preconceived plan? Why those sites? How are the samples taken (ROV arm, push core)? How were samples stored on the ROV, how long before samples reached the surface how are samples processed/stored on-board (significant given the DNA/RNA analysis, e.g. with respect to cross contamination, microbial community shifts)?*: we included more information on the study design, storage and sampling procedure in the material and methods section (see lines 91–102 on the new revised manuscript). We also added a new table (Table 1) with detailed information of the sampling material.

**Detailed response to** *"What material was sampled as 'the samples' (e.g. living coral pieces, coral rubble, sediment with rubble, carbonates)? Size/weight of the samples? Number of samples? Replication?"*: Information about the samples (what is each sample) is detailed in the "Petrography and stable isotopes of carbonates" results (section 3.3). Size of the samples are given with a scale bar in Fig. 7 (A, C, E, F). Weight of the samples was not determined. Each sample is one unit (i. e. coral fragment, carbonate from the based of the Al Gacel MV, carbonate from an active pockmark in Al Gacel MV, and carbonate from the summit of the Al Gacel MV). Replicates used for DNA analysis have been described in section 2.6.1. Furthermore, stable isotopic values obtained from precise sampling sites performed on each sample (section 2.4) are shown in Figure 7 (B, D, F) and Table 3.

**The authors are addressing ecological questions (see e.g. line 34-38, line 50-52 and line 75 "…present-day formation of MDACs and CWCs development…") using studies of carbonates. One of the issues that is particularly relevant for the interpretation of these data is whether the analysis was performed on carbonates with living CWC or not. From the pictures and description, it seems plausible that only dead CWC carbonates were studied (although ln 348 mentions "the necrotic part of living *Madrepora*"), but this begs the question how representative the RNA/DNA/biomarker analysis is when only carbonates of dead CWCs are studied. To what extent do the authors think that the organic components of the carbonates still represent the CWC microbial community? Similarly for the 13C carbonate analysis, is it known well enough whether CWCs leave a distinct isotope mark in the carbonates that is representative for feeding on surface derived organic matter versus hydrocarbons? Targeted sampling of also living CWC pieces and comparison with the sampled carbonates would have provided a means to address this.**

**Response**: since the necrotic coral-carbonate (D03-B1) used for environmental DNA analysis belongs to a living *Madrepora oculata* (see line 303), it is expected that 16S rDNA libraries reveal DNA related to microorganisms related to the corals' microbiota. For instance, sequences related to Enterobacteria and Verrucomicrobia were found in this sample (Supplementary **Table S1**) and are normally in the environment and found associated with corals and other animals (Sorokin et al., 1995; Webster et al., 2016), while *Nitrosococcus* bacteria are ammonia-oxidizers, probably involved in the regulation of nitrogen cycle of the coral's holobiont (Rädecker et al., 2015). Thus, we would have found DNA related to chemosynthetic microorganisms in case the coral fed from the seeping fluids.

Furthermore, it has been supported by many that coral-carbonate skeletons do partially reflect corals nutrition, since part of the $HCO_3^-$ used for its formation comes from the coral's metabolism, i. e. $CO_2$ formed from cellular respiration (Swart, 1983; Zoccola et al., 2015; Nakamura et al., 2018) (lines 392–397 from the revised manuscript). Thus, stable carbon isotopic analysis is an optimal procedure to observe if corals used methane as a carbon source.

**-The authors mention that the ROV had sensors for CO2 and CH4 data and could take NISKIN water samples for CH4. In the results section (ln 219-221 and ln 231) CH4 data are mentioned but in the M&M nothing can be found on sampling location (e.g. height above sediment), sensor calibration, samples handling, sample analyses of the water samples.**

**Response**: Pore-water analysis (from micro-cores) as well as seawater analysis (from Niskin bottles) have been included in this manuscript (see section 2.2.1). However, $CH_4$ measurements have not been included in the material and methods section since those measurements have been done by colleagues from the Subevent-2 project which have previously published the methane values recovered from the Niskin bottles. Sampling procedure can be found in their publication (Sánchez-Guillamón et al., 2015).

**The site description in 3.1 should be partly moved to the Materials and Methods. Only the new results from this study should stay in 3.1.**

**Response**: the Pompeia Province region has been described in detail for the first time in this study. We provide geological structures in this area (Southern and Northern Pompeia Coral ridges, Cold-water Coral Mounds Fields), including novel data (e.g., bathymetry, seismics). Therefore, we consider it appropriate to report these findings in the results sections.

-**The authors infer that "severe seepage results in lethal conditions for CWCs" (line 363 - 364 and 377-378), but I see no evidence for that in the paper. In addition, the authors concluded that CWCs can be entombed by MDAC formation, it is however not clear whether this entombment is the cause of CWC mortality or that this entombment took place after CWC demise following for example from post-glacial decrease in current strength.**

**Response:** We cannot determine if the scleractinian corals embedded in AOM-derived carbonates (samples D10-R3 and D11-R8) were alive or dead when they were buried (lines 812–813 from revised manuscript). However, we observed living corals in areas that are currently affected by seepage (e.g. the Northern Pompeia Coral Ridge, lines 264–265 in the revised manuscript; Fig. 6, C). Furthermore, we observed living octocorals growing on surfaces of currently formed AOM-derived carbonates (e.g., in an active pockmark in the Al Gacel MV, sample D10-R7; Fig. 5, C). These observations indicate that CWCs can live when seepage occurs by means of the "buffer effect" (section 4.3) but severe seepage which cannot be completely buffered may end killing the CWCs.

*Suggestions for minor edits:*
**-ln 48-50: reduce number of refs**
**Response**: done.

**-ln 59: reduce number of refs**
**Response:** done.

**-ln 72-73: reduce number of refs**
**Response:** done.

**-ln 112: Please also give the values of the VPDB used, to avoid confusion**
**Response:** done. Please see lines 140–142 of the new revised manuscript.

**-ln 124: "have a global distribution" instead of "globally widespread"**
**Response:** done in line 21 of the revised manuscript.

**-ln 152: replace "… solid samples were…" with "…sample material was…"**
**Response:** done.

**-ln 230: replace "…by dead.." with "… by shells of the chemosynthetic bivalves Lucinoma…"**
**Response:** done.

**-ln 243: What does "virtually influenced" mean?**
**Response:** "virtually" was deleted.

**-ln 262: "… values ranging from…". From the methods it is unclear on what this range is based, replication, multiple samples?**
**Response:** The range is based on the different values obtained along the same petrographic facies of each sample (Figs. 7 & 9; Table 2). The numbers shown on the petrographic sections of each sample in Figure 7 (Fig. 7, B, D, F), indicate the exact sampling points used for stable isotopic analysis, which values are shown in Table 2. Further information has been included in the foot of Fig. 7 to facilitate this information for the readers.

**-ln 307: What does "proportions" here mean? Do you mean "rates" or "concentrations"?**
**Response:** concentrations. Changed.

**-ln 308: So was methane sampled upon removal of the carbonate blocks?**
**Response:** yes. Information added in line 352 of the new revised manuscript (see Sánchez-Guillamón et al., 2015 for details).

**-ln 368: The authors also mentioned the availability of a CO2 sensor on the ROV. Has this been used to measure aragonite saturation states at the different locations?**
**Response:** Because of the lack of exact data (xcf. Sánchez-Guillamón et al., 2015), aragonite saturation was not calculated. Interestingly, Niskin samples revealed high fCO2 in Al Gacel MV above the seafloor (Sánchez-Guillamón et al., 2015), which may have an effect on the CWC, though experiments showed acclimation of *Lophelia* to changing aragonite saturation (Form et al., 2012). More accurate measurements would have been needed to approach the aragonite saturation state of the different locations.

**-ln 755: Fig 4C. There is a black pointing to "octocorals", but I cannot see these on the picture.**
**Response:** they are on top of the carbonate, difficult to observed since they are semi-transparent. Figure was improved.

**Likewise, please include the response to reviewer 2 point 2o in the discussion.**
**Response**: Response to reviewer 2, point 20 has been included. In the results section we specify the observation of living corals along the Northern Pompeia Coral Ridge (lines 262–264 of revised manuscript; Fig. 6), as well as the presence of living octocorals on top of a currently formed MDAC (line 260 of revised manuscript; Fig. 5). We have also added an aclaration on the foot of Figure 6 (lines 812–813), in which we indicate that we cannot determine if the corals were alive when buried. We have considered that this information is better adapted to those sections, rather than in the discussion section.

**Please ensure that all responses to reviewer 3 are also included in the text.**
**Response**: done.

**Line 53 – 'Supports' should be replaced with 'suggests'.**
**Response**: done.

**Please add a table, referred to in the opening paragraphs of the method section (therefore Table1), detailing study site lat and long, depths, and number of samples of each type collected. Please also indicate the number of replicate samples of each type collected at each location.**
**Response:** A table (now Table 1) has been added to remark and clarify the sampling sites, as well as the type of samples recovered from those sites. Since those samples were unique, there are no replicates of the original samples. Some analysis (e. g. stable isotopes, environmental DNA) do use different replicates from the same sample in order to accomplish stronger results, and those methods can be found in the material and methods section.

**Please add identification of internal and external standards used for GC and isotopic analyses, as well as indications of precision for quantification of lipids and isotopic ratios.**
**Response**: in case of stable isotopic analyses of the carbonates, accuracy and reproducibility were checked through the replicate analysis of a standard (NBS19), and the reproducibility was better than 0.1 ‰. This information is already provided in the methods section.

In case of stable carbon isotopic analyses of organic compounds, $CO_2$ of known stable carbon isotopic composition was used for internal calibration. This information is already provided in the methods section. The reference $CO_2$ was calibrated with a standard (IAEA600). Standard deviations of duplicate sample measurements were better than 1.0 ‰. We included this information into the method section.

Lipid biomarkers were not quantified, therefore no standard was needed.

**Line 413-end of discussion. Your hypothesis regarding a biological buffer requires further discussion and possibly evidence. The two questions that occur to me are: 1) Is the presence of sulphide and methane normally prohibitive to the existence of CWCs? At what concentrations do they become problematic? Sulphide is of course toxic at certain concentrations, but non-chemosynthetic 'normal' or 'background' benthic fauna can and do inhabit sites with some level of sulphide flux (see Bell et al. 2016, Frontiers in Marine Science), and methane is even less of a problem. 2) Do you have evidence (i.e. porewater and bottom water methane and S- concentrations) to show that bacterial activity does indeed lead to reductions in sulphide concentrations such that they allow colonisation by CWCs? I'd suggest that there is another explanation, which is that CWCs are tolerant to some extent of sulphide**

**and methane fluxes, however sulphide and methane may cause some degree of stress, which may at least partially explain the poor health (low abundance of living material) that you observed.**

**Detail response to** *"1) Is the presence of sulphide and methane normally prohibitive to the existence of CWCs? At what concentrations do they become problematic? Sulphide is of course toxic at certain concentrations, but non-chemosynthetic 'normal' or 'background' benthic fauna can and do inhabit sites with some level of sulphide flux (see Bell et al. 2016, Frontiers in Marine Science), and methane is even less of a problem":* We agree that non-chemosynthetic fauna is able to live in conditions where sulfide and methane fluxes are present in "some level". Interestingly, when seepage of methane and/or sulfide occurs, there is normally chemosynthetic-fauna related to this seepage, which are actually "buffering" the harmful "levels" that could affect those non-chemosynthetic fauna if they would not feed on the seeped fluids. As we observed in Fig. 12, A, which represents the active pockmark found in the Al Gacel MV (Fig. 5), CWCs are living in an active pockmark and actually colonizing a currently-formed AOM carbonate. Furthermore, methane is indeed not toxic for CWCs, but its emission decreases pH and complicates carbonate precipitation (which affects CWCs like scleractinians).

**Detailed response to** *"2) Do you have evidence (i.e. porewater and bottom water methane and S-concentrations) to show that bacterial activity does indeed lead to reductions in sulphide concentrations such that they allow colonisation by CWCs? I'd suggest that there is another explanation, which is that CWCs are tolerant to some extent of sulphide and methane fluxes, however sulphide and methane may cause some degree of stress, which may at least partially explain the poor health (low abundance of living material) that you observed":* we have now included S- and Fe values obtained from pore-water and seawater samples (see section 2.2.1 and lines 269 – 275 of new revised manuscript). S- and Fe values in the pore-water are higher that those from the bottom seawater, which indicates its consumption. This can be explained by the observation of framboidal pyrite inside the carbonate D10-R7 (Fig. 8 C–D), as well as environmental bacterial DNA sequences which indicate the presence of sulfide-oxidizing bacteria. Furthermore, ROV images also indicate the presence of siboglinid worms that also consume this sulfide.

**Authors additional modifications**

1) Line 3: Francisco Javier González instead of Javier González.

2) Lines 4 and 10: Names and information related to new co-authors, Esther Santofimia and Enrique López-Pamo.

3) Line 35: "such as Siboglinidae worms" added.

4) Line 59: "in northern Rockall Trough".

5) Line 70: "with only a few living corals".

6) Line 118: new section including water analysis.

7) Lines 268–275: water parameters improved.

8) Lines 360–361: water analysis results included in discussion

[revised manuscript text omitted]

cubic inches) with a total of 860 cubic inches. The obtained data were recorded with an active streamer (SIG®16.3x40.175; 150 m length with 3 sections of 40 hydrophones each). The shot interval was 6 seconds and the recording length 5 seconds two-way travel time (TWT). Data processing (filtering and stacking) was performed on board with Hot Shots software.

**2.2. Video survey and analysis**

A remotely operated vehicle (ROV-6000 Luso, operated by EMEPC) was used for photographic documentation (high definition digital camera, 1024x1024 pixel) and sampling. The ROV was further equipped with a STD/CTD-

SD204 sensor (*in-situ* measurements of salinity, temperature, oxygen, conductivity, sound velocity and depth),

HydroC$^{TM}$ sensors (*in-situ* measurements of $CO_2$ and $CH_4$), and Niskin bottles (CH$_4$ concentrations, pH and redox potential measurements)s (CH$_4$ concentrations), and a ROV core sampler (up to 16 cm).

**2.2.1. Seawater and pore-water analysis**

Niskin water-samples and micro-cores covering the water/sediment interface were recovered from an active pockmark close to the summit of the Al Gacel MV (D10-N4, D10-C5, D10-C8; same site as carbonate-sample

D10-R7) as well as directly from its summit (D11-N9, D11-C10). Redox potentials (ORP) and pH-values of the water contained in the Niskin bottles were measured on site with HANNA portable instruments (HI 9025). Pore- water from the micro-cores was immediately extracted by centrifuging 10 cm thick slices of the sediments. Upon extraction, the pore-water was filtered with syringe filters of cellulose acetate (0.2 μm pore), acidified with distilled nitric acid ($HNO_3$), and stored under 4 °C before further analysis. Major and trace elements were subsequently measured with an Agilent 7500c inductively coupled plasma mass spectrometer (ICP-MS). Method accuracy and precision was checked by external standards (MIV, EPA, NASC, CASS). The precision was better than 5 % RSD

(residual standard deviation) and the accuracy better than 4%. Concentrations of $S^{2-}$ were measured with a Hanch-

Lange DR 2800 spectrophotometer (cuvette test kit LCK 653).

One Niskin water-sample (D10-N4) and two micro-cores (D10-C5 and D10-C8) were recovered from active pockmark close to the summit of the Al Gacel MV (as carbonate D10-R7). Likewise, one Niskin water-sample (D11-N9) and one micro-core (D11-C10) were recovered from the summit of the Al Gacel MV. Redox potential (ORP) and pH were measured on site from Niskin water-samples with HANNA portable instruments (HI 9025).

[revised manuscript text omitted]


[revised manuscript text omitted]
. ~~Furthermore, soluble S$^{2-}$ value of 0.23 μM and Fe value of 1.74 μM were measured in pore-water of shallow cores sampled in the same area (D10-C8; **Table 2**). These values are higher than the concentration measured in seawater (D10 N4; **Table 2**), indicating sulfide is being produced by the activity of AOM and also being consumed before it reaches the water column as well as the reduced iron. At the same time, abundant framboidal pyrite in the carbonate (**Fig. 8, C–D**) and SRB-related DNA (**Fig. 11**) evidences microbial sulfate reduction in the environment. All these data clearly demonstrate that the carbonates have been formed via AOM, fueled by fluids from the underlying mud diapir.~~

Other carbonate samples from the Al Gacel MV (i.e. D10-R3 and D11-R8) probably have also been formed due to AOM as they are  isotopically depleted as well (δ$^{13}$C values between ca. −25 and −15 ‰, **Fig. 9**, **Table 3**). However, no active gas bubbling was observed during sampling, even though both samples still contain open voids which could form pathways for  fluids. Several characteristics of these voids (e.g. dark halos formed by pyrite, brownish margins due to organic matter enrichments) are very similar to those of methane-derived carbonate conduits (cf. Reitner et al., 2015). This could imply that the intensity of hydrocarbon-rich seepage and consequently AOM, may have fluctuated through time. The relatively low dominance of crocetane and PMI in a carbonate sampled from the summit of Al Gacel MV (D11-R8; **Fig. 10**. The moderately depleted δ$^{13}$C values of crocetane/phytane and PMI in this sample (−57.2 ‰ and −74.3 ‰, respectively; **Table 4**) could be due to mixing effects and are thus also in agreement with varying intensities of AOM in the environment. The  presence of only few AOM-related DNA sequences (**Fig. 11**) and partly oxidized pyrites in the carbonate D10-R3 from the base of the Al Gacel MV (**Fig. 8, A–B**) are well in line with this scenario.

There is no evidence for eruptive extrusions of muddy materials at the coral ridges. In the Southern Pompeia Coral Ridge (**Fig. 3**), diapirs appear to rather promote an upward migration of hydrocarbon-rich fluids in a divergent way throughout a more extensive seabed area. This results in a continuous and diffused seepage, which promotes the occurrence of AOM and the formation of MDACs at the base of the ridges, related to the sulphate-methane transition zone (SMTZ) related to the sulphate-methane transition zone (SMTZ) (Boetius et al., 2000; Hinrichs and Boetius, 2002; González et al., 2012a). This is in good accordance with the detection of methane (80 – 83 nM)

at the Northern Pompeia Coral Ridge and the presence of sulfide-oxidizing bacterial mats and shells of dead chemosynthetic bivalves at the western part of the ridge (**Fig. 6, A**). Likewise, the CWC Mounds Field surrounding the Southern Pompeia Coral Ridge (**Fig. 3**) is thoroughly characterized by micro-seeps, due to ascending fluids from OPs through low-angle faults. This type of focused seepage may promote formation of MDAC pavements in deeper layers of the sediments (**Fig. 3**), similar to coral ridges along the Pen Duick Escarpment (Wehrmann et al.,

2011). The generation of MDAC-hotspots at sites of such seepage also explain the geometry of the downward tapering cones (**Fig. 3**).

**4.2. Ecological meaning of hydrocarbon-rich seepage for CWCs**

Our data suggests contemporaneous micro-seepage and CWC growth in the Pompeia Province (e.g. **Fig. 4, B**).

This relationship has also been observed elsewhere, e.g. in the North Sea and off Mid Norway (Hovland, 1990;

Hovland & Thomsen, 1997), and the Angola margin (Le Guilloux et al., 2009). Corals utilize $HCO_3^-$ deriving from both the environment and the internal production of $CO_2$ for skeleton biomineralization (Swart, 1983; Zoccola et al., 2015; Nakamura et al., 2018). Hence, a potential utilization of methane as a carbon source should be reflected in the $\delta^{13}C$ signatures of their skeletons. However, scleractinian fragments recovered from the Al Gacel MV

(embedded in carbonates D10-R3 and D11-R8, from the base and summit of the volcano, respectively) and the

Northern Pompeia Coral Ridge (D03-B1, necrotic part of a living *Madrepora oculata*) displayed barely depleted

$\delta^{13}C$ values (ca. −8 to −1 ‰; **Fig. 9**; **Table 23**), close to the $\delta^{13}C$ of marine seawater (0 ± 3 ‰, e.g. Hoefs, 2015).

These values do not support Since it has been proposed that corals utilize $HCO_3^-$ deriving from both the environment and the internal production of $CO_2$ for skeleton biomineralization (Swart, 1983; Zoccola et al., 2015;

Nakamura et al., 2018), we expect $\delta^{13}C$ values in corals' carbonates to reflect whether or not they use methane as a carbon source. In our study, $\delta^{13}C$ values This doesdo 
[revised manuscript text omitted]

---

## Author Response (AR3)

**Authors response to Editor (part 2)**

We are thankful for your supportive comments and we hope we have successfully addressed your questions and concerns.

*Specific Comments from Editor*

**Firstly, Table 1 is very enlightening. Could you please confirm for me, are your conclusions about the C sources utilized by the corals based only on sample D03-B1, or do they also use data from other carbonate samples, which elsewhere in the manuscript are also referred to as coral skeleton?**

**Response:** Our conclusions addressing the unlikely use of methane as carbon source by CWCs is based on DNA data (section 3.5) from sample D03-B1 (a necrotic fragment from a living Madrepora oculata) and stable carbon isotopes also from sample D03-B1, and from embedded corals in samples D10-R3 and D11-R8. Results concerning stable carbon isotopes are found in section 3.3 (lines 275–304 of revised manuscript; Table 4; Figs. 7 & 9). We have additionally included this information again in Table 1 (lines 737–738 of revised manuscript)

**Secondly, I do not feel that my comment about your proposed biological buffer has been fully answered. I am happy with the statement that 'These microbes may form a biological buffer...', however I am not happy with the stronger statement starting 'This model explains the observed co-existence of ...' I do not feel that other explanations (such as coral tolerance of sulphide etc.) have been fully discussed. Please either moderate and shorten section 4.3 (it would be acceptable if limited mostly to the first paragraph), or provide substantial additional argument, support from the literature, and description of patterns in your own data which support the hypothesis.**

**Response:** we fully understand your concerns and revised the chapter 4.3 accordingly, essentially following your suggestions to moderate and shorten the chapter. We furthermore tried to emphasize that the proposed "biological buffers" appear to be a further, additional ecological factor that is relevant for CWC development in the study area. We also stress that CWCs have certain ecological capabilities that may allow them to thrive at seepage-influenced localities and provide relevant references. Finally, we tried to stress that the geographical extent of these biological buffers has to be further evaluated (i.e., local vs. regional vs. global relevance). In combination, our changes hopefully make clear that the biological buffers are an important aspect for CWCs in seepage-influences environments, but that this by no means exclude other influences such as ecological capabilities of the corals or other environmental factors.

**Authors response to Editor**

*Dear Dr Rincón-Tomás*

*Thank you for submitting your revised manuscript. I would like to request some additional changes, many of which aim to ensure that the reasoning you provide in response to reviewer comments is actually included in the manuscript discussion. Please could you therefore undertake further revisions to accommodate the comments below.*

*Best regards,*

*Clare Woulds*

**Authors**: we appreciate your constructive comments on our manuscript. We are also thankful for the extra time you have given us to improve the manuscript and address successfully all discussion points.

**You mention twice that the aim of the study was to '...address the linkage between CWCs and present day formation of MDACs.' The fact that two reviewers have questioned the study objectives / hypotheses supports my feeling that the phrase 'address the linkage' is not sufficiently explicit. Please re-phrase your aim and research questions in plainer language, and state the hypothesis that you were testing.**
**Response:** we have now added some more sentences indicating the hypothesis of our study, which to prove if CWCs are non-chemosynthetic organisms or they rather harbor chemosynthetic symbionts which allow them consuming some of the seeped fluids.

**Please ensure that the answer to reviewer 2 point 19 is included in the discussion, with appropriate acknowledgement that the tissue you analysed was not living biomass (i.e. coral polyps), that analysis of such live tissue would be required to draw firm conclusions that methane C was not a major dietary C source, and stressing that the conclusion that can be drawn is that methane C was not a major C source during building of the exoskeleton. I recognise your point that if the corals were using methane derived C, then when it was metabolised some of it may be incorporated into the exoskeleton. However, the lack of (much) evidence for this is a rather tenuous way of drawing a conclusion about how the corals fulfilled their metabolic needs.**
**Response**: We have added more information addressing this issue in the discussion, between lines 391–394 and 409 from the revised manuscript. We specify our analyzed sample is a "necrotic part of a living *Madrepora oculata*" in line 396.

**Likewise, please include the response to reviewer 2 point 2o in the discussion.**

**Response**: Response to reviewer 2, point 20 have been included in the results (lines 262–263) and in the foot of Figure 6 (lines 812–813). We have considered that this information is better adapted to those sections, rather than in the discussion section.

**Please ensure that all responses to reviewer 3 are also included in the text.**
**Response**: done.

**Line 53 – 'Supports' should be replaced with 'suggests'.**
**Response**: done.

**Please add a table, referred to in the opening paragraphs of the method section (therefore Table1), detailing study site lat and long, depths, and number of samples of each type collected. Please also indicate the number of replicate samples of each type collected at each location.**
**Response:** A table (now Table 1) has been added to remark and clarify the sampling sites, as well as the type of samples recovered from those sites. Since those samples were unique, there are no replicates of the original samples. Some analysis (e. g. stable isotopes, environmental DNA) do use different replicates from the same sample in order to accomplish stronger results, and those methods can be found in the material and methods section.

**Please add identification of internal and external standards used for GC and isotopic analyses, as well as indications of precision for quantification of lipids and isotopic ratios.**
**Response**: in case of stable isotopic analyses of the carbonates, accuracy and reproducibility were checked through the replicate analysis of a standard (NBS19), and the reproducibility was better than 0.1 ‰. This information is already provided in the methods section.

In case of stable carbon isotopic analyses of organic compounds, $CO_2$ of known stable carbon isotopic composition was used for internal calibration. This information is already provided in the methods section. The reference $CO_2$ was calibrated with a standard (IAEA600). Standard deviations of duplicate sample measurements were better than 1.0 ‰. We included this information into the method section.

Lipid biomarkers were not quantified, therefore no standard was needed.

**Line 413-end of discussion. Your hypothesis regarding a biological buffer requires further discussion and possibly evidence. The two questions that occur to me are: 1) Is the presence of sulphide and methane normally prohibitive to the existence of CWCs? At what concentrations do they become problematic? Sulphide is of course toxic at certain concentrations, but non-chemosynthetic 'normal' or 'background' benthic fauna can and do inhabit sites with some level of sulphide flux (see Bell et al. 2016, Frontiers in Marine Science), and methane is even less of a problem. 2) Do you have evidence (i.e. porewater and bottom water methane and S- concentrations) to show that bacterial activity does indeed lead to reductions in sulphide concentrations such that they allow colonisation by CWCs? I'd suggest that there is another explanation, which is that CWCs are tolerant to some extent of sulphide**

**and methane fluxes, however sulphide and methane may cause some degree of stress, which may at least partially explain the poor health (low abundance of living material) that you observed.**

**Detail response to** *"1) Is the presence of sulphide and methane normally prohibitive to the existence of CWCs? At what concentrations do they become problematic? Sulphide is of course toxic at certain concentrations, but non-chemosynthetic 'normal' or 'background' benthic fauna can and do inhabit sites with some level of sulphide flux (see Bell et al. 2016, Frontiers in Marine Science), and methane is even less of a problem":* We agree that non-chemosynthetic fauna is able to live in conditions where sulfide and methane fluxes are present in "some level". Interestingly, when seepage of methane and/or sulfide occurs, there is normally chemosynthetic-fauna related to this seepage, which are actually "buffering" the harmful "levels" that could affect those non-chemosynthetic fauna if they would not feed on the seeped fluids. As we observed in Fig. 12, A, which represents the active pockmark found in the Al Gacel MV (Fig. 5), CWCs are living in an active pockmark and actually colonizing a currently-formed AOM carbonate. Furthermore, methane is indeed not toxic for CWCs, but its emission decreases pH and complicates carbonate precipitation (which affects CWCs like scleractinians).

**Detailed response to** *"2) Do you have evidence (i.e. porewater and bottom water methane and S-concentrations) to show that bacterial activity does indeed lead to reductions in sulphide concentrations such that they allow colonisation by CWCs? I'd suggest that there is another explanation, which is that CWCs are tolerant to some extent of sulphide and methane fluxes, however sulphide and methane may cause some degree of stress, which may at least partially explain the poor health (low abundance of living material) that you observed":* we have now included S- and Fe values obtained from pore-water and seawater samples (see section 2.2.1 and lines 269 – 274 of new revised manuscript). S- and Fe values in the pore-water are higher that those from the bottom seawater, which indicates its consumption. This can be explained by the observation of framboidal pyrite inside the carbonate D10-R7 (Fig. 8 C–D), as well as environmental bacterial DNA sequences which indicate the presence of sulfide-oxidizing bacteria. Furthermore, ROV images also indicate the presence of siboglinid worms that also consume this sulfide.

**Authors response to Referee n° 1**

We are thankful for the constructive and helpful comments that have helped us to improve our manuscript. We are aware that the manuscript holds a high amount of data which can be difficult to follow at some points and tried to keep it as concise as possible. We considered all comments carefully and modified and followed most of the suggestions.

*Specific Comments from Referee n° 1*

**2) The introduction reads well. One question is whether you have a testable hypothesis. Are you trying to ask whether the corals are fueled by fluids versus scavenging from currents. How are you going to distinguish between mechanisms?**

Response: the aim of the study is to address the linkage between CWCs and present day formation of MDACs in the Pompeia Province. For this purpose, we combined analyses of ROV images, geophysical data and sample materials. For instance, we analyzed $\delta^{13}C$ signatures of coral skeletons to evaluate whether these organisms were directly relying on $CH_4$. We found that the coral skeletons exhibited significantly higher $\delta^{13}C$ values than the co-occurring AOM-derived carbonates, thus not supporting $CH_4$ as important carbon source. Rather, the corals were feeding on material suspended in currents.

**3) In the methods please add section in which you describe the Experimental Design. How many samples were collected and from where? The descriptions of the laboratory methods are okay. However, I have no idea if you sampled thoroughly enough.**

Response: we included more detailed information on our sample strategy and study design in the material and methods section.

**4) In Table 2, will readers know what Identifier means? I realize that the numbers correspond to pictures in the figures. However, it is very confusing to have to put the figure next to the table to interpret the data in the table. There must be a better way to present the data.**

Response: done. We replaced "Identifier" by "Identification number in Fig. 7". In Addition, we added an additional column to the table in which we provide information on the analyzed material.

**5) Rather than using code numbers for the sampling sites, it would help readers if you used descriptive names, such as 'active seep', etc.**

Response: done. We have revised the use of code numbers throughout the manuscript.

**6) Although amplicon sampling for microbial group is okay. Do you have evidence for microbial growth and activity? Perhaps in the discussion indicate which samples come from fresh material and are likely to have fresh DNA versus samples in which the DNA could be old and preserved. I realized this is inferred by looking at the pictures, but again this is a convoluted way to present a story.**

Response: we have improved the information concerning the DNA material related to each sample in the manuscript, and we have specified the type of sample from which the DNA has been extracted (lines 180–183 in the revised manuscript). Furthermore, we added some extra information in Fig. 11 to clarify and remain the type of sample. DNA analyses cannot conclude if DNA is "old" or "fresh", but we can estimate (together with other analyses) if the sample used for this analysis is fresh or not. but we can infer this by assessing the relative age and preservation of the analyzed sample. For instance, an AOM-derived carbonate recovered from an active pockmark (sample D10-R7) exhibits more DNA of AOM-related microorganisms (ANME and SRB) than oxidized AOM-derived carbonates recovered from regions that are currently not affected by seepage (sample D10-R3).

**7) I suppose the model is okay. However, again a better presentation of the data might lead readers to the conclusion rather than relying on the author's story.**
**Response**: done. We have modified the last paragraph of the section 4.3. for a better understanding of our model (lines 438–445 in the revised manuscript).

*Technical Comments from Referee n° 1*
**1) Line 19: consider saying, 'rate a seepage via focused, scattered, diffused, etc.'**
**Response**: done. We revised the sentence to "the type of seepage such as focused, scattered, diffused or eruptive".

**2) Line 34: change 'which' to 'that'.**
**Response**: done.

**3) Line 36: change to 'typically, they thrive, etc.'**
**Response**: done.

**4) Line 45: change 'ecological' to 'environmental' and 'are discussed to control' to 'influence'.**
**Response**: done.

**5) Line 51: delete 'e.g.'.**
**Response**: done.

**6) Line 53: change 'e.g.' to 'for example'.**
**Response**: done.

**7) Line 65: delete 'i.e.' and the parentheses. The text is not an example rather it is the description of 'coral graveyards.'**
**Response**: done.

**Authors response to Referee n° 2**

We are thankful for your constructive feedback and the helpful comments. We have considered and addressed your suggestions carefully, and almost all have been followed in the revised manuscript.

**Detail Comments from Referee n° 2**

**1) Line 1. Title. The text after the hyphen: 'living on the edge' is unnecessary and adds nothing to the title. What edge? I suggest removing this.**

**Response:** we would like to keep the text "living on the edge" to emphasize that hydrocarbon-rich seepage has both advantages and disadvantages for cold-water corals growth.

**2) Lines 26-27. Abstract Delta C13 values of the coral skeletons (see below)**

**Response:** see discussion on reviewer comment n° 19 below.

**3) Line 31. Abstract. Suggest 'seeping' rather than 'seeped' fluids.**

**Response:** done.

**4) Line 61. Suggest 'In addition' to replace 'On the other hand', as this is not a contrasting observation.**

**Response:** done.

**5) Line 76. 'Englobes' is not an English word. Seems like a transliteration of 'encompasses'.**

**Response:** done.

**6) Line 128. Don't start sentence with a number – spell it out.**

**Response:** done.

**7) Line 152. Can the authors give a little more detail of the nature of the samples used for the DNA work. Are these MDACs?**

**Response:** done. We now provide more information on the nature of the samples (lines 182–185 in the revised manuscript).

**8) Lines 192-195. The background information about the Gulf of Cadiz isn't really results and would go better at the start of section 2.**

**Response:** we agree that the background information of the Gulf of Cádiz is not part of results. However, the Pompeia Province region, which our study is focused on, has not been described in detail so far. We here provide the first description of geological structures in this area (Southern and Northern Pompeia Coral ridges, Cold-water Coral Mounds Fields), including novel data (e.g., bathymetry, seismics). For this reason, we consider it appropriate to report these findings in the results sections.

**9) Line 241 and other places. It's quite difficult at the moment to correlate the isotopic**

data in Table 2 with the sample points in Figure 7, because the specimen images in
Figure 7 are not quite large enough to distinguish samples of authigenic carbonates from
embedded coral skeletons. Therefore, could the authors add a column into Table 2 that makes it
clear what the samples are for each of the isotopic data points, e.g. authigenic carbonate or coral
skeleton.

**Response:** done. One more column has been added in Table 2 as proposed, indicating the type of samples
from which stable isotopic analyses are.

**10) Line 253. Replace 'stems' with 'comes'.**
**Response**: done.

**11) Line 254. In the figure the 'worms' look like serpulid worm tubes. Is this so? In which
case please add this information.**
**Response**: done.

**12) Line 291. Replace 'On the contrary' with 'In contrast'.**
**Response:** done.

**13) Line 296. Spell out '2D' at start of sentence.**
**Response:** done.

**14) Line 305 and elsewhere. What is 'dripping-like' seepage? This isn't a description I
recognize, so it would be helpful if the authors specify what this means.**
**Response**: done. "Dripping-like refers to intermittent bubbling fluids" (lines 342–343 in the revised
manuscript).

**15) Line 317. Suggest 'data', rather than 'evidences'.**
**Response**: done.

**16) Line 330. I'm unclear where is being referred to here.**
**Response**: removed.

**17) Line 332. 'appear', not 'appears', as preceding diapirs is plural.**
**Response:** done.

**18) Line 339. Typo. Angle not angel.**
**Response**: done.

**19) Lines 346-354. The authors here suggest that the seawater-like values of the delta C13 from the
dead scleractinian skeletons and those embedded in the MDAC show that the corals do not use**

**methane as a food source, either directly or through symbionts. The authors need to be careful here, because some seep organisms that demonstrably do use methane (and sulfide) from seep fluids for food via endosymbionts produce carbonate skeletons that also have seawater-like delta C13 signatures. I am referring here to vesicomyid and bathymodiolin bivalves, that sequester seawater bi-carbonate ions to produce their shells. Using this model, having seawater-like delta C13 values in the coral skeletons does not prove that these animals do not use chemosynthetic food sources at the site. Really, to be able to settle this conclusively, authors would have to do isotopic, histological and DNA work on living corals from their site, not just on skeletal material and MDAC. In addition, it would be worth noting that scleractinian corals are found embedded in ancient seep carbonates too (see Goedert and Peckmann 2005); there may be some useful comparative isotopic data in that paper.**

**Response:** We included the paper by Goedert and Peckmann, 2005. We fully agree that analyses of coral tissues ($\delta^{13}C$, DNA) would add important information on their nutrition and metabolic relationships. However, we still regard $\delta^{13}C$ values of their skeletons as valuable proxy for the possible uptake of $CH_4$. Corals utilize $HCO_3^-$ deriving from both the environment and the internal production of $CO_2$ for skeleton biomineralization (Swart, 1983; Zoccola et al., 2015; Nakamura et al., 2018). Therefore, if they uptake $CH_4$ as a carbon source, the $CO_2$ produced from $CH_4$ metabolism would be used, and consequently parts of the $HCO_3^-$ utilized for biomineralization would be isotopically depleted. This "mixing effect" would result in at least partially depleted $\delta^{13}C$ values of the skeletons, similar to some chemosynthetic vesicomyid and lucinid bivalves (Hein et al., 2006). The skeletons of the corals analyzed herein, however, exhibit significantly higher $\delta^{13}C$ values than the co-occurring AOM-derived carbonates. Thus, they are not indicative for $CH_4$ as important carbon source.

**20) Lines 364-367. The entombment of coral skeletons by MDAC may have no consequence to corals, if they are already dead. It's not entirely clear from the text if the corals associated with the MDAC are dead or alive. If they are alive then this argument is stronger. Also, in most seep environments MDACs form in the subsurface where AOM reactions are occurring. Is this the case at this site? What proof is there of active MDAC formation at the sediment-water interface, as indicated in Figure 12? This is pertinent to the arguments in section 4.3.**

**Response:** We cannot determine if the scleractinian corals embedded in AOM-derived carbonates (samples D10-R3 and D11-R8) were alive or dead when they were buried (lines 812-813 in the revised manuscript). However, we observed living corals in areas that are currently affected by seepage (e.g. the Northern Pompeia Coral Ridge, lines 262–263 in the revised manuscript; Fig. 6, C). Furthermore, we observed living octocorals growing on surfaces of currently formed AOM-derived carbonates (e.g., in an active pockmark in the Al Gacel MV, sample D10-R7; Fig. 5, C). These observations imply that corals in these regions are directly affected by methane seepage and the microbially mediated formation of carbonates due to AOM.

**Detailed response to** *"What material was sampled as 'the samples' (e.g. living coral pieces, coral rubble, sediment with rubble, carbonates)? Size/weight of the samples? Number of samples?*

*Replication?"*: Information about the samples (what is each sample) is detailed in the "Petrography and stable isotopes of carbonates" results (section 3.3). Size of the samples are given with a scale bar in Fig. 7 (A, C, E, F). Weight of the samples was not determined. Each sample is one unit (i. e. coral fragment, carbonate from the based of the Al Gacel MV, carbonate from an active pockmark in Al Gacel MV, and carbonate from the summit of the Al Gacel MV). Replicates used for DNA analysis have been described in section 2.6.1. Furthermore, stable isotopic values obtained from precise sampling sites performed on each sample (section 2.4) are shown in Figure 7 (B, D, F) and Table 2.

**The authors are addressing ecological questions (see e.g. line 34-38, line 50-52 and line 75 "…present-day formation of MDACs and CWCs development…") using studies of carbonates. One of the issues that is particularly relevant for the interpretation of these data is whether the analysis was performed on carbonates with living CWC or not. From the pictures and description, it seems plausible that only dead CWC carbonates were studied (although ln 348 mentions "the necrotic part of living *Madrepora*"), but this begs the question how representative the RNA/DNA/biomarker analysis is when only carbonates of dead CWCs are studied. To what extent do the authors think that the organic components of the carbonates still represent the CWC microbial community? Similarly for the 13C carbonate analysis, is it known well enough whether CWCs leave a distinct isotope mark in the carbonates that is representative for feeding on surface derived organic matter versus hydrocarbons? Targeted sampling of also living CWC pieces and comparison with the sampled carbonates would have provided a means to address this.**

**Response**: since the necrotic coral-carbonate (D03-B1) used for environmental DNA analysis belongs to a living *Madrepora oculata* (see line 302), it is expected that 16S rDNA libraries reveal DNA related to microorganisms related to the corals' microbiota. For instance, sequences related to Enterobacteria and Verrucomicrobia were found in this sample (Supplementary **Table S1**) and are normally in the environment and found associated with corals and other animals (Sorokin et al., 1995; Webster et al., 2016), while *Nitrosococcus* bacteria are ammonia-oxidizers, probably involved in the regulation of nitrogen cycle of the coral's holobiont (Rädecker et al., 2015). Thus, we would have found DNA related to chemosynthetic microorganisms in case the coral fed from the seeping fluids.

Furthermore, it has been supported by many that coral-carbonate skeletons do partially reflect corals nutrition, since part of the $HCO_3^-$ used for its formation comes from the coral's metabolism, i. e. $CO_2$ formed from cellular respiration (Swart, 1983; Zoccola et al., 2015; Nakamura et al., 2018) (lines 393–397 from the revised manuscript). Thus, stable carbon isotopic analysis is an optimal procedure to observe if corals used methane as a carbon source.

**-The authors mention that the ROV had sensors for CO2 and CH4 data and could take NISKIN water samples for CH4. In the results section (ln 219-221 and ln 231) CH4 data are mentioned but in the M&M nothing can be found on sampling location (e.g. height above sediment), sensor calibration, samples handling, sample analyses of the water samples.**

**Response**: Pore-water analysis (from micro-cores) as well as seawater analysis (from Niskin bottles) have been included in this manuscript (see section 2.2.1). However, $CH_4$ measurements have not been included in the material and methods section since those measurements have been done by colleagues from the Subevent-2 project which have previously published the methane values recovered from the Niskin bottles. Sampling procedure can be found in their publication (Sánchez-Guillamón et al., 2015).

**The site description in 3.1 should be partly moved to the Materials and Methods. Only the new results from this study should stay in 3.1.**

**Response**: the Pompeia Province region has been described in detail for the first time in this study. We provide geological structures in this area (Southern and Northern Pompeia Coral ridges, Cold-water Coral Mounds Fields), including novel data (e.g., bathymetry, seismics). Therefore, we consider it appropriate to report these findings in the results sections.

**-The authors infer that "severe seepage results in lethal conditions for CWCs" (line 363 - 364 and 377-378), but I see no evidence for that in the paper. In addition, the authors concluded that CWCs can be entombed by MDAC formation, it is however not clear whether this entombment is the cause of CWC mortality or that this entombment took place after CWC demise following for example from post-glacial decrease in current strength.**

**Response:** We cannot determine if the scleractinian corals embedded in AOM-derived carbonates (samples D10-R3 and D11-R8) were alive or dead when they were buried (lines 812–813 from revised manuscript). However, we observed living corals in areas that are currently affected by seepage (e.g. the Northern Pompeia Coral Ridge, lines 258–259 in the revised manuscript; Fig. 6, C). Furthermore, we observed living octocorals growing on surfaces of currently formed AOM-derived carbonates (e.g., in an active pockmark in the Al Gacel MV, sample D10-R7; Fig. 5, C). These observations indicate that CWCs can live when seepage occurs by means of the "buffer effect" (section 4.3) but severe seepage which cannot be completely buffered may end killing the CWCs.

*Suggestions for minor edits:*

**-ln 48-50: reduce number of refs**

**Response**: done.

**-ln 59: reduce number of refs**

**Response:** done.

**-ln 72-73: reduce number of refs**

**Response:** done.

**-ln 112: Please also give the values of the VPDB used, to avoid confusion**

**Response:** done. Please see lines 139–140 of the new revised manuscript.

**-ln 124: "have a global distribution" instead of "globally widespread"**

**Response:** done in line 16 of the revised manuscript.

**-ln 152: replace "… solid samples were…" with "…sample material was…"**
**Response:** done.

**-ln 230: replace "…by dead.." with "… by shells of the chemosynthetic bivalves Lucinoma…"**
**Response:** done.

**-ln 243: What does "virtually influenced" mean?**
**Response:** "virtually" was deleted.

**-ln 262: "… values ranging from…". From the methods it is unclear on what this range is based, replication, multiple samples?**
**Response:** The range is based on the different values obtained along the same petrographic facies of each sample (Figs. 7 & 9; Table 2). The numbers shown on the petrographic sections of each sample in Figure 7 (Fig. 7, B, D, F), indicate the exact sampling points used for stable isotopic analysis, which values are shown in Table 2. Further information has been included in the foot of Fig. 7 to facilitate this information for the readers.

**-ln 307: What does "proportions" here mean? Do you mean "rates" or "concentrations"?**
**Response:** concentrations. Changed.

**-ln 308: So was methane sampled upon removal of the carbonate blocks?**
**Response:** yes. Information added in line 347 of the new revised manuscript (see Sánchez-Guillamón et al., 2015 for details).

**-ln 368: The authors also mentioned the availability of a CO2 sensor on the ROV. Has this been used to measure aragonite saturation states at the different locations?**
**Response:** Because of the lack of exact data (xcf. Sánchez-Guillamón et al., 2015), aragonite saturation was not calculated. Interestingly, Niskin samples revealed high $fCO_2$ in Al Gacel MV above the seafloor (Sánchez-Guillamón et al., 2015), which may have an effect on the CWC, though experiments showed acclimation of *Lophelia* to changing aragonite saturation (Form et al., 2012). More accurate measurements would have been needed to approach the aragonite saturation state of the different locations.

**-ln 755: Fig 4C. There is a black pointing to "octocorals", but I cannot see these on the picture.**
**Response:** they are on top of the carbonate, difficult to observed since they are semi-transparent. Figure was improved.

**References**

[revised manuscript text omitted]